# Don't Reinvent the Wheel, Just Realign the Spokes: Resource-Efficient Federated Fine-Tuning via Rank-Wise Expert Assembly

Yebo Wu [* 1]   Jingguang Li [* 2]   Zhijiang Guo [3 4]   Li Li [1]

## Abstract

Federated fine-tuning presents a promising avenue for adapting Large Language Models (LLMs) to downstream tasks while preserving data privacy. However, the prohibitive computational and communication overhead of LLM adaptation inhibits its deployment on resource-constrained edge devices. In this paper, we propose SMARTFED, a resource-efficient framework that circumvents expensive training from scratch by intelligently reusing knowledge embedded in existing LoRA modules. To fully exploit this potential and ensure scalability, we introduce the Mixture of Rank-Wise Experts (MoRE). MoRE decomposes LoRA modules into fine-grained rank-level experts, which are selectively activated based on input semantics and resource budgets. Furthermore, to optimize resource utilization, we propose Elastic Expert Quota Allocation (EEQA), a strategy that adaptively distributes expert capacity across parameter matrices based on their contribution to model performance. Extensive evaluations across multiple benchmarks demonstrate that SMARTFED significantly outperforms state-of-the-art methods in both model performance and training efficiency. Our code is publicly available at https://github.com/benmagnifico/SmartFed.

## 1. Introduction

Large Language Models (LLMs) (Yang et al., 2025; Liu et al., 2024) have demonstrated remarkable capabilities across diverse domains (Wu et al., 2026c; Xu et al., 2025; 2026c;b;d;a), yet their potential is often bottlenecked by the need for task-specific alignment (Tian et al., 2024). While fine-tuning is effective, downstream data is frequently fragmented across edge devices and bound by strict privacy regulations (e.g., GDPR) (McMahan et al., 2017; Wu et al., 2025b; 2024; Zhang et al., 2025; 2026), rendering centralized training infeasible. Federated fine-tuning (Wu et al., 2026d;b) has emerged as a compelling solution, enabling collaborative adaptation while preserving data privacy. However, a significant hurdle remains: the sheer scale of LLMs makes full-parameter fine-tuning prohibitively expensive for resource-constrained edge devices.

To bridge this gap, various parameter-efficient federated fine-tuning methods have been proposed (Bian et al., 2025), with LoRA (Hu et al., 2022) garnering significant attention due to its efficiency. Yet, a critical inefficiency persists in current paradigms: existing methods typically train LoRA modules from scratch (Zhang et al., 2024; Wu et al., 2025a) (Figure 1(a)). This process demands hundreds of communication rounds to converge, incurring substantial computation and transmission costs. Meanwhile, the open-source community has amassed a wealth of LoRA modules fine-tuned on diverse tasks (Huang et al., 2024a). This presents a paradoxical landscape where we possess abundant blocks of knowledge but continue to manufacture new ones from raw materials. This observation prompts a fundamental question: *Can we simply reuse existing LoRA modules to adapt LLMs to new tasks, thereby circumventing costly retraining?*

In this paper, we propose SMARTFED, a resource-efficient federated fine-tuning framework that shifts the paradigm from *training* to *composing*. Instead of learning parameters from scratch, SMARTFED intelligently reuses the knowledge embedded in existing, publicly available LoRA modules. Specifically, the server retrieves task-relevant modules from repositories (e.g., LoRAHub (Huang et al., 2024a)), and a lightweight, trainable router dynamically activates them based on input semantics (Figure 1(b)). Consequently, edge devices only need to optimize the router, drastically reducing local training burdens and communication overhead.

However, directly fusing entire LoRA outputs is coarse-grained, and as more modules are retrieved, computational overhead also increases. To overcome this, we introduce the Mixture of Rank-Wise Experts (MoRE). MoRE decomposes each LoRA module along the rank dimension,

---

[*]Equal contribution [1]University of Macau [2]KAIST [3]HKUST [4]HKUST (Guangzhou). Correspondence to: Zhijiang Guo <zhijiangguo@hkust-gz.edu.cn>, Li Li <llili@um.edu.mo>.

*Proceedings of the 43rd International Conference on Machine Learning*, Seoul, South Korea. PMLR 306, 2026. Copyright 2026 by the author(s).

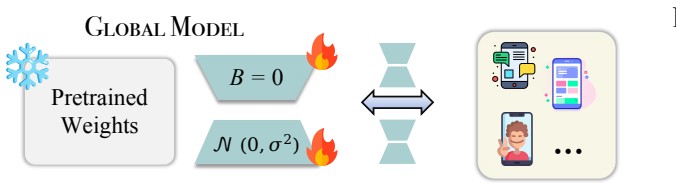

(a) Classic Federated Fine-Tuning Framework.

(b) Our SmartFed Framework.

*Figure 1.* Illustration of the classic federated fine-tuning framework versus our SMARTFED. Classic methods train LoRA modules from scratch (a), while SMARTFED updates only the router (b), substantially reducing resource overhead.

treating the paired vectors from the $\mathbf{A}$ and $\mathbf{B}$ matrices as independent, fine-grained experts. This architectural decomposition allows the model to "realign the spokes" of the wheel—selectively activating only the most relevant rank-level features to construct a high-precision adaptation. Furthermore, recognizing that different parameter matrices contribute unevenly to model performance, we propose Elastic Expert Quota Allocation (EEQA). This strategy adaptively assigns the budget of activated experts to different matrices based on their contribution, ensuring that computing resources are concentrated where they matter most.

Extensive evaluations demonstrate the superiority of SMARTFED, achieving up to 13.82% average performance gains, $3.95\times$ faster convergence, $31.47\times$ lower communication overhead, and $3.61\times$ reduced energy consumption. Moreover, SMARTFED exhibits strong data efficiency; for instance, on math-word problems, it outperforms baselines by 8.02% while using only 10% of the training data, highlighting its value in data-scarce domains.

## 2. Background and Motivation

### 2.1. Limitations of Existing LoRA Reuse Methods

A prevailing approach for reusing task-adapted LoRA modules is to synthesize a unified module via Linear Arithmetic (Ilharco et al., 2023). For instance, Zhang et al. (2023a) perform element-wise addition over corresponding parameter matrices (Figure 2(a)). Formally, given $N$ LoRA modules $\{(\mathbf{B}_n, \mathbf{A}_n)\}_{n=1}^{N}$ fine-tuned on $N$ distinct tasks, the composite module is defined as:

$$\mathbf{B}_{\text{add}} = \sum_{n=1}^{N} \lambda_n \mathbf{B}_n, \quad \mathbf{A}_{\text{add}} = \sum_{n=1}^{N} \lambda_n \mathbf{A}_n, \quad (1)$$

where $\lambda_n$ is a weighting coefficient controlling each module's contribution. While computationally efficient, this approach imposes a rigid constraint: all modules must share identical dimensions. More critically, it introduces substantial mathematical noise, particularly when constituent modules originate from semantically conflicting tasks.

Consider two LoRA modules $(\mathbf{B}_1, \mathbf{A}_1)$ and $(\mathbf{B}_2, \mathbf{A}_2)$ with

residual updates $\Delta\mathbf{h}_1 = \mathbf{B}_1\mathbf{A}_1\mathbf{x}$ and $\Delta\mathbf{h}_2 = \mathbf{B}_2\mathbf{A}_2\mathbf{x}$. Ideally, an effective knowledge integration should yield:

$$\begin{aligned} \mathbf{h}' &= \mathbf{W}_0\mathbf{x} + \lambda_1\Delta\mathbf{h}_1 + \lambda_2\Delta\mathbf{h}_2 \\ &= \mathbf{W}_0\mathbf{x} + \lambda_1\mathbf{B}_1\mathbf{A}_1\mathbf{x} + \lambda_2\mathbf{B}_2\mathbf{A}_2\mathbf{x}. \end{aligned} \quad (2)$$

However, under the naive parameter merging defined in Equation (1), the actual operation becomes:

$$\begin{aligned} \mathbf{h}' &= \mathbf{W}_0\mathbf{x} + (\lambda_1\mathbf{B}_1 + \lambda_2\mathbf{B}_2)(\lambda_1\mathbf{A}_1 + \lambda_2\mathbf{A}_2)\mathbf{x} \\ &= \mathbf{W}_0\mathbf{x} + \underbrace{\lambda_1^2\mathbf{B}_1\mathbf{A}_1\mathbf{x} + \lambda_2^2\mathbf{B}_2\mathbf{A}_2\mathbf{x}}_{\text{Target Task Adaptation}} \\ &\quad + \underbrace{\lambda_1\lambda_2\mathbf{B}_1\mathbf{A}_2\mathbf{x} + \lambda_1\lambda_2\mathbf{B}_2\mathbf{A}_1\mathbf{x}}_{\text{Destructive Cross-Task Interference}}. \end{aligned} \quad (3)$$

As shown above, the merged parameters generate cross-task terms (e.g., $\mathbf{B}_1\mathbf{A}_2$). These terms deviate from the task-specific manifolds, causing destructive interference. To mitigate this algebraic interference, the Mixture of LoRA Experts (MoLE) (Huang et al., 2024b) adopts an output-level aggregation strategy, treating each LoRA module as an independent expert (Figure 2(b)). While MoLE circumvents parameter conflict, it introduces two critical bottlenecks.

- **Coarse Adaptation.** MoLE activates the *entire* LoRA module, neglecting the heterogeneous contributions of individual rank-one components within a module. To quantify this limitation, we compare MoLE against a fine-grained method, our proposed Mixture of Rank-Wise Experts (MoRE), on LLaMA2-7B across skill-composition tasks (Chinese+Math, Chinese+Code). As shown in Figure 3(a), while MoLE outperforms Linear Arithmetic by avoiding cross-term noise, it still lags behind the fine-grained MoRE by up to 5.01%. This performance gap underscores that treating LoRA modules as atomic units fails to fully exploit the subtle, rank-level knowledge embedded within them.

- **Scalability Barrier.** MoLE incurs a linear increase in computational overhead as more modules are retrieved. We highlight this in Figure 3(b), which plots per-sample inference latency against the number of

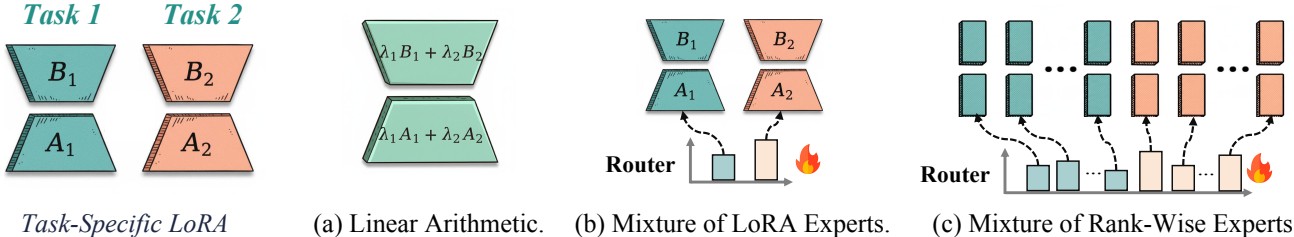

*Figure 2.* Comparison of knowledge reuse strategies. (a) **Linear Arithmetic** merges parameters via element-wise arithmetic; (b) **MoLE** aggregates outputs from entire modules; (c) **MoRE** (Ours) dynamically integrates fine-grained outputs from rank-wise components.

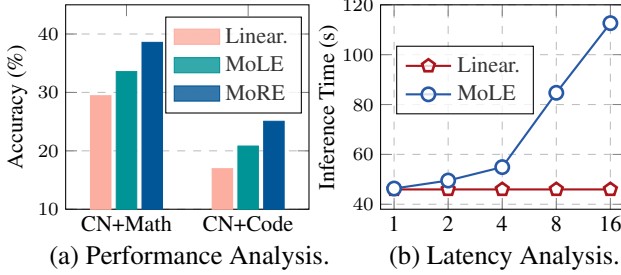

*Figure 3.* Quantitative analysis of knowledge reuse strategies. (a) Performance comparison across different tasks. (b) Inference latency comparison under varying numbers of LoRA modules.

LoRA modules. While Linear Arithmetic maintains constant cost, MoLE's latency spikes dramatically (e.g., $1.84\times$ with eight modules). This prohibits scalable deployment on resource-constrained edge devices.

These results expose the limitations of existing methods: Linear Arithmetic is efficient but noisy, while MoLE is coarse and expensive. This motivates the design of our Mixture of Rank-Wise Experts (Section 3.1), which offers an efficient and fine-grained framework for knowledge reuse.

### 2.2. Heterogeneous Importance of Rank-Wise Experts

While MoRE enables fine-grained knowledge reuse, full expert activation remains computationally prohibitive for edge devices, necessitating sparse selection. However, a critical question remains: *how should we effectively allocate this limited expert budget?* We hypothesize that rank-wise experts from different LoRA modules contribute unevenly to model performance, and that allocating expert capacity accordingly can improve overall accuracy.

To verify this hypothesis, we define a quantitative metric for expert importance. For a rank-wise expert $E_m$, we measure the extent to which it alters the input representation. Given an input $\mathbf{h} \in \mathbb{R}^d$ and the expert's output $\mathbf{h}_{E_m} = E_m(\mathbf{h})$, the importance score $s_{E_m}$ is defined as:

$$s_{E_m} = 1 - \cos(\mathbf{h}, \mathbf{h}_{E_m}) = 1 - \frac{\mathbf{h}^\top \mathbf{h}_{E_m}}{||\mathbf{h}|| \cdot ||\mathbf{h}_{E_m}||}, \quad (4)$$

where $\cos(\cdot)$ denotes cosine similarity. Higher scores imply more significant feature transformations, indicating greater expert importance. Figure 4 reveals striking disparities in expert importance across three dimensions:

- **Cross-Task Variation (Figure 4(a)):** Experts derived from different tasks (Chinese vs. Math) exhibit distinct score distributions, with Math experts showing a higher mean importance (0.0169 vs. 0.0143).

- **Matrix-Type Sensitivity (Figure 4(b)):** The contribution of experts varies by matrix type. For instance, in the first layer, experts in the Value matrix have higher importance scores compared to the Query matrix.

- **Layer-Specific Difference (Figure 4(c)):** Expert importance is not uniform across layers; rather, it exhibits significant distributional shifts between layers (e.g., the distinct patterns observed in Layer 2 versus Layer 1).

Crucially, this heterogeneity translates directly to model performance. As shown in Figure 5, we observe a sharp asymmetry in performance gains: increasing the expert quota for Query yields a substantial accuracy jump (30.58% → 36.98%), whereas an equivalent increase for Value results in diminishing returns (34.07%). These results confirm that a uniform budget allocation is inefficient, motivating the design of our Elastic Expert Quota Allocation (Section 3.2).

## 3. Our Method: SMARTFED

### 3.1. Mixture of Rank-Wise Experts

To enable effective knowledge reuse, we propose the Mixture of Rank-Wise Experts (MoRE). The core insight of MoRE is that a LoRA module is not an atomic monolith, but rather an aggregate of independent rank-one components. By decomposing modules along the rank dimension, MoRE transforms coarse-grained LoRA modules into a flexible pool of fine-grained, lightweight experts.

**Rank-Wise Decomposition.** Mathematically, the standard LoRA update $\Delta\mathbf{W}\mathbf{x} = \mathbf{B}\mathbf{A}\mathbf{x}$ can be reformulated as a

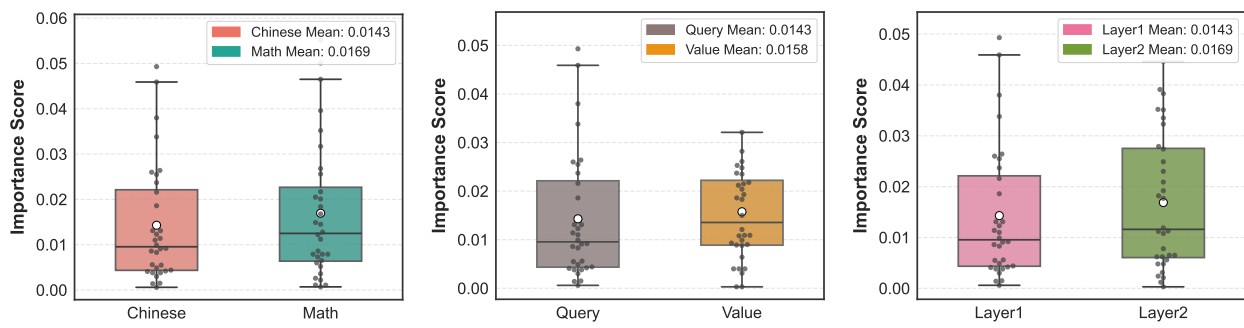

*(a)* Expert importance across tasks.     *(b)* Expert importance across matrix types.     *(c)* Expert importance across layers.

*Figure 4.* Heterogeneity of rank-wise expert importance in LLaMA2-7B on the Chinese mathematical reasoning task. (a) Comparison of the first-layer `Query` matrix across Chinese and Math LoRA modules. (b) Comparison between `Query` and `Value` matrices in the first layer (Chinese LoRA). (c) Layer-wise comparison of the `Query` matrix (Layer 1 vs. Layer 2, Chinese LoRA).

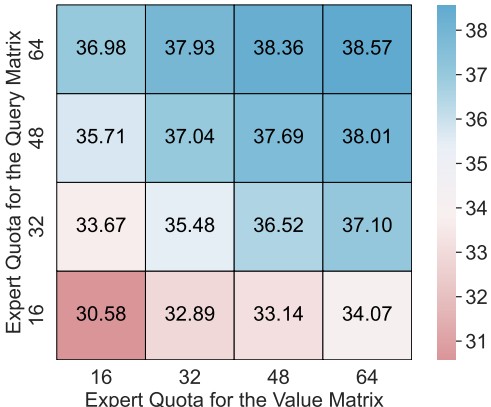

*Figure 5.* Impact of expert quota allocation on model performance.

summation of independent rank-one projections:

$$\Delta\mathbf{W}\mathbf{x} = \sum_{i=1}^{r}\mathbf{B}_{:,i}\left(\mathbf{A}_{i,:}\mathbf{x}\right), \tag{5}$$

where $r$ is the LoRA rank, and $\mathbf{B}_{:,i} \in \mathbb{R}^{d\times 1}$ and $\mathbf{A}_{i,:} \in \mathbb{R}^{1\times d}$ denote the $i$-th column of $\mathbf{B}$ and row of $\mathbf{A}$, respectively. We treat each term $\mathbf{B}_{:,i}\mathbf{A}_{i,:}$ as an independent *rank-wise expert*, formally defined as: $E_i(\mathbf{x}) = (\mathbf{B}_{:,i}\mathbf{A}_{i,:})\mathbf{x}$, where $i \in [1, r]$. This decomposition allows us to access and reuse knowledge at the subspace level.

**Construction and Routing.** Figure 6 illustrates the MoRE architecture. The server constructs an expert pool by decomposing $N$ LoRA modules into $M = \sum_{n=1}^{N} r_n$ rank-wise components. To orchestrate this pool, we employ a trainable router for input-conditioned activation. Specifically, given an input token $\mathbf{x}$, the router computes gating scores $\mathbf{g}$:

$$\mathbf{g} = \text{softmax}(\mathbf{W}_{\text{router}}\mathbf{x}), \tag{6}$$

where $\mathbf{W}_{\text{router}} \in \mathbb{R}^{M\times d}$ is the only learnable parameter matrix during adaptation. To ensure scalability on resource-

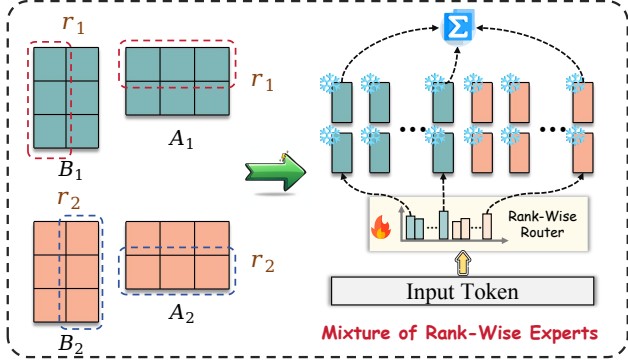

*Figure 6.* Overview of the Mixture of Rank-Wise Experts (MoRE).

constrained devices, we enforce sparsity by activating only the top-$K$ experts (where $K \ll M$). This is implemented via a binary mask $\mathbf{m} \in \{0,1\}^{M}$:

$$\tilde{\mathbf{g}} = \mathbf{g} \odot \mathbf{m}, \quad \text{where } \mathbf{m}_m = \mathbb{I}(\mathbf{g}_m \in \text{Top-}K(\mathbf{g})). \tag{7}$$

Therefore, the final output aggregates the output of the base parameter matrix with the weighted contributions of the selected experts:

$$\mathbf{h}' = \mathbf{W}_0\mathbf{x} + \sum_{m=1}^{M}\tilde{\mathbf{g}}_m \cdot E_m(\mathbf{x}). \tag{8}$$

**Paradigm Shift.** MoRE fundamentally shifts the federated fine-tuning paradigm from parameter training to expert selection, significantly reducing resource overhead.

### 3.2. Elastic Expert Quota Allocation

Motivated by the observations in Section 2.2, we propose Elastic Expert Quota Allocation (EEQA), which adaptively determines the number of activated experts for each parameter matrix based on its contribution to model performance. The federated workflow with EEQA operates in two stages:

**Stage 1: Initialization and Profiling.** Prior to adaptive training, this stage initializes the federated architecture and computes the expert contributions.

1. **Assembly and Distribution:** The server first retrieves task-relevant LoRA modules and assembles them into the MoRE (as detailed in Section 3.1). The global model is then distributed to all participating devices.

2. **Importance Profiling:** Before training commences, devices calculate the importance score for each expert via Equation (4). These local scores are uploaded to the server to establish the matrix importance profile.

3. **Initial Assignment:** The server randomly samples a subset of devices and distributes the router to them. Initially, a uniform expert quota, denoted by $K$, is allocated to each matrix based on hardware resources.

**Stage 2: Iterative Adaptive Training.** The core of EEQA lies in its ability to refine allocations throughout training.

1. **Local Update and Feedback.** Selected devices train the router using their local private data. After local training, each device calculates the cumulative importance score $S_j^{\text{Sum}}$ for each parameter matrix $j \in [1, J]$ by aggregating the scores of its activated experts:

$$S_j^{\text{Sum}} = \sum_{m \in \mathcal{I}_j} \tilde{\mathbf{g}}_m \cdot s_{E_m}, \tag{9}$$

where $\mathcal{I}_j$ denotes the index set of rank-wise experts associated with the $j$-th parameter matrix. The updated router and the set of importance scores $\{S_j^{\text{Sum}}\}_{j=1}^{J}$ are transmitted back to the server.

2. **Global Aggregation and Reallocation.** The server aggregates the local router updates and importance scores via FedAvg (McMahan et al., 2017). It subsequently applies a softmax function to derive the normalized allocation weights $\alpha_j$:

$$\alpha_j = \frac{\exp(S_j^{\text{Sum}})}{\sum_{j'=1}^{J} \exp(S_{j'}^{\text{Sum}})}, \quad j = 1, \dots, J. \tag{10}$$

The expert budget is redistributed using the **EEQA** (Algorithm 1). This process consists of two phases: *Phase 1* ensures a base quota allocation proportional to importance, while *Phase 2* greedily distributes any residual budget to the most critical matrices.

3. **Continuous Evolution.** The server then distributes the updated router and adaptive quotas to a new client subset. This iterative cycle enables the system to progressively channel computational resources toward the "spokes" (rank-wise experts) that contribute most to model performance, continuing until convergence.

---

**Algorithm 1** The Elastic Expert Quota Allocation Strategy.

**Require:** Normalized importance scores $\{\alpha_j\}_{j=1}^{J}$, expert budget $B = K \cdot J$, and per-matrix upper bound $M$.
**Ensure:** Expert quota for each matrix: $\{\tilde{q}_j\}_{j=1}^{J}$.

1:   **Phase 1: Proportional Guarantee**
2:   **for** $j = 1$ **to** $J$ **do**
3:     $\tilde{q}_j \leftarrow \min\left(\lfloor \alpha_j \cdot B \rfloor, M\right)$ {Allocate base share}
4:   **end for**
5:   $R \leftarrow B - \sum_{j=1}^{J} \tilde{q}_j$ { Calculate Residual Budget }
6:   **Phase 2: Greedy Residual Distribution**
7:   $\mathcal{S} \leftarrow \{j \mid \tilde{q}_j < M\}$ {Identify matrices with capacity}
8:   Sort $\mathcal{S}$ by $\alpha_j$ in descending order
9:   **for** $j \in \mathcal{S}$ **do**
10:     **if** $R = 0$ **then**
11:       **break**
12:     **end if**
13:     $\Delta \leftarrow \min(M - \tilde{q}_j, R)$
14:     $\tilde{q}_j \leftarrow \tilde{q}_j + \Delta; \quad R \leftarrow R - \Delta$
15:   **end for**

16:   **Return** $\{\tilde{q}_j\}_{j=1}^{J}$

---

## 4. Theoretical Analysis

We provide theoretical justification for the effectiveness of SMARTFED. Our analysis focuses on: (1) how SMARTFED suppresses task-irrelevant noise and algebraically eliminates cross-task cross-terms, and (2) the bounded rounding loss of the Elastic Expert Quota Allocation (EEQA) strategy relative to a capped proportional allocation target.

### 4.1. Noise Suppression and Interference Removal

Existing methods indiscriminately utilize all LoRA parameters, which can introduce (1) *task-irrelevant noise accumulation* and (2) *algebraic cross-task cross-terms* from conflicting parameter combinations (e.g., $\mathbf{B}_1 \mathbf{A}_2$). We formalize how SMARTFED addresses both issues.

**Theorem 4.1** (Noise Suppression and Algebraic Cross-Term Elimination). *Let $\mathcal{M}$ denote the set of all rank-wise experts for a given parameter matrix, with total count $M = |\mathcal{M}| = \sum_{n=1}^{N} r_n$. For an input $\mathbf{x}$, let $\mathcal{S}_{\text{irr}} \subset \mathcal{M}$ be the task-irrelevant experts ($|\mathcal{S}_{\text{irr}}| = \rho M$). Assume expert contributions contain additive zero-mean noise $\xi_i$ satisfying $\mathbb{E}[\xi_i] = 0$ and $\mathbb{E}\|\xi_i\|^2 = \sigma^2$ for $i \in \mathcal{S}_{\text{irr}}$, and noises are uncorrelated between experts and independent of the routing weights. Define the **dense all-expert baseline** noise vector as $\Phi_{\text{dense}} := \sum_{i \in \mathcal{S}_{\text{irr}}} \xi_i$, and the SMARTFED noise vector as $\Phi_{\text{SMARTFED}} := \sum_{i \in \mathcal{S}_{\text{irr}}} \hat{g}_i \xi_i$, where $\{\hat{g}_i\}_{i=1}^{M}$ are the masked gating weights with $\hat{g}_i = 0$ for inactive experts outside the activated set $\mathcal{A}$ ($|\mathcal{A}| = K$). If the uninformative routing baseline is exchangeable with respect to expert indices (e.g., the activated set $\mathcal{A}$ is selected uniformly at ran-*

*dom and the squared gating mass is assigned symmetrically across experts), and $S_2 := \sum_{i=1}^{M} \hat{g}_i^2$ denotes the realized squared-weight mass, then conditioned on $S_2$, the expected squared noise norm satisfies:*

$$\mathbb{E}\big[\|\Phi_{\text{SMARTFED}}\|^2 \mid S_2\big] = \frac{S_2}{M} \cdot \mathbb{E}\big[\|\Phi_{\text{dense}}\|^2\big] = \rho\, S_2\, \sigma^2. \tag{11}$$

*Equivalently, after taking expectation over the routing randomness, we have*

$$\mathbb{E}\big[\|\Phi_{\text{SMARTFED}}\|^2\big] = \rho\, \sigma^2\, \mathbb{E}[S_2]. \tag{12}$$

*Furthermore, SMARTFED **algebraically eliminates** the cross-task interference terms $\mathbf{B}_n \mathbf{A}_{n'}$ $(n \neq n')$ inherent to parameter merging via subspace knowledge aggregation.*

*Proof.* The proof is deferred to Appendix A.1. □

**Remark.** Equation (11) bridges the gap between theoretical baselines and practical implementations by isolating the effect of weight normalization $S_2$:

- **Unit-weight (Linear Arithmetic counterpart):** $\hat{g}_i = 1 \implies S_2 = K$.

- $\ell_2$**-normalized:** $\hat{g}_i = 1/\sqrt{K} \implies S_2 = 1$.

- **Softmax-then-mask (Our Implementation):** Since $\sum \hat{g}_i \leq 1$, we have $S_2 = \sum \hat{g}_i^2 \leq (\sum \hat{g}_i)^2 \leq 1$.

This implies that our Softmax-based implementation achieves a noise bound no worse than the $\ell_2$-normalized theoretical case. Note that the uninformative routing assumption provides an average-case baseline; in practice, the learned router can further reduce noise when it assigns less squared gating mass to $\mathcal{S}_{\text{irr}}$ than the uninformative baseline.

### 4.2. Rounding Guarantee of EEQA

EEQA allocates an integer quota $q_j \in \{0, 1, \ldots, M\}$ to each matrix based on importance weights $\alpha_j$ under a total budget $B$ and per-matrix cap $M$. We analyze EEQA as an approximation to an *ideal proportional target* $q_j^\star = \min(\alpha_j B,\ M)$, which corresponds to a capped proportional allocation before enforcing integrality. To measure the quality of a quota allocation, we use $f(q)$ to denote the normalized utility obtained by assigning $q$ rank-wise experts to a parameter matrix, capturing its expected contribution to representation improvement or downstream performance.

**Proposition 4.2** (Additive Guarantee to the Proportional Target). *Let $f$ be a non-decreasing, 1-Lipschitz function on $[0, M]$. Let $\tilde{q}$ be the allocation produced by Algorithm 1. Then EEQA achieves the following guarantee:*

$$\sum_{j=1}^{J} \alpha_j f(\tilde{q}_j) \geq \sum_{j=1}^{J} \alpha_j f(q_j^\star) - 1. \tag{13}$$

*Proof.* The proof is deferred to Appendix A.2. □

Proposition 4.2 shows that the only loss relative to the ideal capped proportional target comes from integer rounding. In particular, the additive constant 1 is independent of $J$ and $B$, and can be interpreted as the maximum utility loss incurred by "missing" at most one unit of quota due to integrality. The proportional target $q_j^\star$ prevents the starvation of critical components; by closely tracking $q_j^\star$, EEQA ensures that high-importance matrices receive adequate capacity despite discrete constraints and per-matrix caps.

## 5. Experiments

### 5.1. Experimental Setup

Following prior works (Wang et al., 2024a), we evaluate SMARTFED on three LLMs: LLaMA2-7B, LLaMA2-13B (Touvron et al., 2023), and Qwen2-7B (Bai et al., 2023). Our experiments cover three skill-composition tasks: Chinese mathematical reasoning, Chinese code generation, and hard math–word problems. For baselines without LoRA knowledge reuse, we train LoRA from scratch on Math23K (Wang et al., 2017), DoIT (Song et al., 2025), and MathCodeInstruct (Wang et al., 2024b). Evaluation is performed on MGSM (Shi et al., 2023), DoIT, and GSM-Hard (Gao et al., 2023). Appendix B provides the acquisition process for skill-specific LoRA modules, while Appendix C presents additional implementation details.

### 5.2. Baselines

**1) Knowledge-Free Methods**: These approaches train LoRA from scratch, including FedIT (Zhang et al., 2024), FwdLLM (Xu et al., 2024b), DoFIT (Xu et al., 2024a), and FedAdapter (Cai et al., 2023). We further adapt AdaLoRA (Zhang et al., 2023b), a parameter budget allocation method, to the federated scenario.

**2) Knowledge-Reuse Methods:** Due to the absence of federated LoRA reuse baselines, we adapt several established centralized methods to the federated setting, including Linear Arithmetic (Zhang et al., 2023a), LoRAHub (Huang et al., 2024a), MoLE (Huang et al., 2024b), and LoRA-Flow (Wang et al., 2024a). Detailed descriptions of these methods are provided in Appendix D.

### 5.3. Performance Evaluation

As summarized in Table 1, SMARTFED significantly surpasses all baselines across diverse skill-composition tasks. On LLaMA2-7B, SMARTFED outperforms knowledge-free methods with gains of up to 4.51%, 7.63%, and 8.02% across the three tasks, underscoring the benefit of reusing LoRA knowledge. However, naive combinations of LoRAs remain inadequate: Linear Arithmetic and LoRAHub suffer

*Table 1.* Performance comparison of different methods on diverse skill-composition tasks across LLMs. *Activated Rank* denotes the average number of activated ranks per parameter matrix. The best results are highlighted in **bold**.

| LLM | Method | MGSM | DoIT | GSM-Hard | Average | Activated Rank |
|-----|--------|------|------|----------|---------|----------------|
| **LLaMA2-7B** | FedIT | 30.97 | 44.96 | 56.17 | 44.03 (↓ 6.72) | 32 |
| | FwdLLM | 32.35 | 47.18 | 57.61 | 45.71 (↓ 5.04) | 32 |
| | DoFIT | 31.28 | 46.09 | 56.89 | 44.75 (↓ 6.00) | 32 |
| | FedAdapter | 32.47 | 48.70 | 58.05 | 46.41 (↓ 4.34) | 32 |
| | AdaLoRA | 33.69 | 50.13 | 59.15 | 47.66 (↓ 3.09) | 32 |
| | Linear Arith. | 29.43 | 43.81 | 54.90 | 42.71 (↓ 8.04) | 32 |
| | LoRAHub | 33.41 | 49.25 | 58.79 | 47.15 (↓ 3.60) | 32 |
| | MoLE | 33.56 | 49.54 | 58.69 | 47.26 (↓ 3.49) | 64 |
| | LoRA-Flow | 33.75 | 49.95 | 59.06 | 47.59 (↓ 3.16) | 64 |
| | **SMARTFED** | **35.48** | **52.59** | **64.19** | **50.75** | 32 |
| **LLaMA2-13B** | FedIT | 43.68 | 55.00 | 63.87 | 54.18 (↓ 8.77) | 32 |
| | FwdLLM | 47.11 | 56.92 | 66.58 | 56.87 (↓ 6.08) | 32 |
| | DoFIT | 45.85 | 56.37 | 64.91 | 55.71 (↓ 7.24) | 32 |
| | FedAdapter | 48.29 | 58.57 | 66.79 | 57.88 (↓ 5.07) | 32 |
| | AdaLoRA | 50.91 | 60.41 | 68.84 | 60.05 (↓ 2.90) | 32 |
| | Linear Arith. | 42.24 | 52.79 | 63.19 | 52.74 (↓ 10.21) | 32 |
| | LoRAHub | 50.43 | 60.06 | 68.42 | 59.64 (↓ 3.31) | 32 |
| | MoLE | 50.20 | 60.17 | 68.36 | 59.58 (↓ 3.37) | 64 |
| | LoRA-Flow | 51.12 | 60.32 | 68.70 | 60.05 (↓ 2.90) | 64 |
| | **SMARTFED** | **53.18** | **63.01** | **72.65** | **62.95** | 32 |
| **QWEN2-7B** | FedIT | 30.59 | 44.17 | 55.04 | 43.27 (↓ 7.08) | 32 |
| | FwdLLM | 31.82 | 46.41 | 56.52 | 44.92 (↓ 5.43) | 32 |
| | DoFIT | 30.74 | 45.39 | 55.90 | 44.01 (↓ 6.34) | 32 |
| | FedAdapter | 31.99 | 48.00 | 57.21 | 45.73 (↓ 4.62) | 32 |
| | AdaLoRA | 33.05 | 49.45 | 58.01 | 46.84 (↓ 3.51) | 32 |
| | Linear Arith. | 29.10 | 43.05 | 53.79 | 41.98 (↓ 8.37) | 32 |
| | LoRAHub | 33.04 | 48.30 | 57.74 | 46.36 (↓ 3.99) | 32 |
| | MoLE | 33.11 | 48.79 | 57.81 | 46.57 (↓ 3.78) | 64 |
| | LoRA-Flow | 33.19 | 49.23 | 58.24 | 46.89 (↓ 3.46) | 64 |
| | **SMARTFED** | **35.31** | **52.11** | **63.62** | **50.35** | 32 |

*Figure 7.* Overhead analysis of different federated fine-tuning methods on LLaMA2-7B for the Chinese+Math task.

average performance drops of 8.04% and 3.60% relative to SMARTFED, respectively. These results suggest that direct parameter merging inevitably induces task-irrelevant noise.

While MoLE and LoRA-Flow mitigate this issue through dynamic expert routing, their coarse-grained integration still incurs average performance drops of 3.49% and 3.16% relative to SMARTFED. Beyond this, SMARTFED exhibits remarkable generalization across model scales and archi-

tectures. On LLaMA2-13B, its advantage becomes even more pronounced, yielding average gains of up to 10.21%. Comparable improvements are also observed on Qwen2-7B, where SMARTFED achieves up to 8.37% average gains.

## 5.4. Efficiency Evaluation

We then evaluate the efficiency of SMARTFED along three dimensions: *training time*, *communication cost*, and *energy*

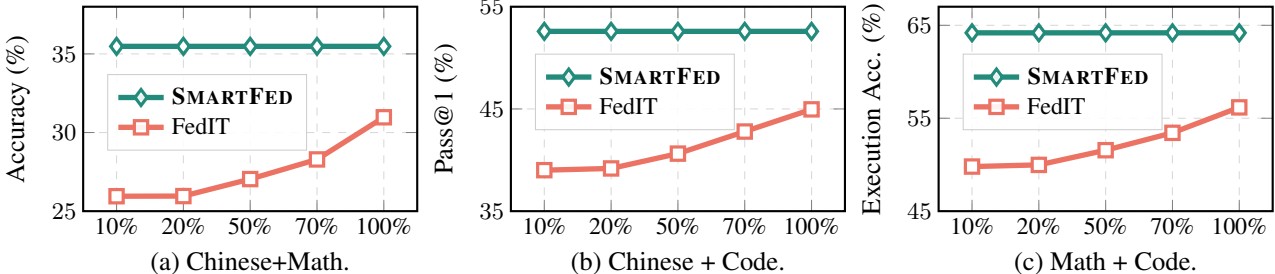

*Figure 8.* Performance comparison under varying fractions of training data on LLaMA2-7B across three downstream tasks.

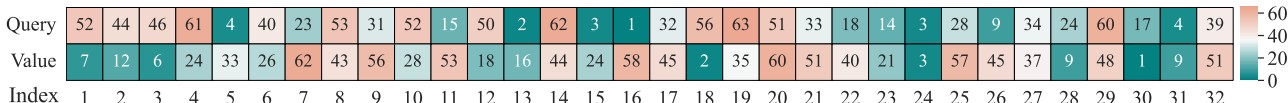

*Figure 9.* Distribution of expert quotas across different layers of LLaMA2-7B for the Chinese+Math task.

*consumption.* All experiments are conducted on NVIDIA H800 GPUs, with energy consumption measured using `CodeCarbon` (Patterson et al., 2021). Figure 7 shows that SMARTFED significantly reduces resource overhead, accelerating model convergence by up to 3.95×, lowering communication cost by up to 31.47×, and reducing energy consumption by up to 3.61×. These improvements arise from SMARTFED 's reuse of task-adapted LoRA modules, which avoids costly retraining. Consequently, SMARTFED significantly enhances system efficiency, making federated fine-tuning practical in resource-constrained environments.

### 5.5. Additional Analysis

**Data Efficiency.** We evaluate the data efficiency of SMARTFED on LLaMA2-7B under varying proportions of training data. As shown in Figure 8, SMARTFED consistently outperforms FedIT across all data regimes. Notably, with only 10% of the training data, SMARTFED surpasses FedIT trained on the full (100%) dataset by 4.51%, 7.63%, and 8.02% on the three tasks. These results underscore the exceptional data efficiency of SMARTFED, a property particularly valuable in data-scarce domains such as healthcare.

**Quota Allocation Analysis.** To understand the role of EEQA, we analyze the distribution of expert quotas across parameter matrices. As shown in Figure 9, the quota allocations exhibit significant variations across layers and matrix types. For example, the 16th-layer `Value` matrix is allocated 58 experts, whereas the corresponding `Query` matrix is assigned only one. This disparity demonstrates EEQA's ability to accurately identify and prioritize critical experts, thereby optimizing knowledge utilization.

**Fine-Grained Knowledge Fusion.** To understand MoRE's effectiveness, we visualize the average fusion weights of rank-wise components using a sample from the Chi-

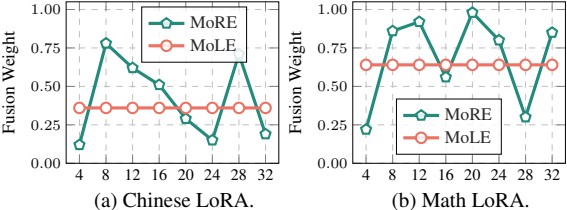

*Figure 10.* Average fusion weights for the `Query` matrix in the third layer of LLaMA2-7B on the Chinese+Math task.

*Table 2.* Performance evaluation on recent state-of-the-art models.

| Method | Evaluation Task | | | Average |
|---|---|---|---|---|
| | **MGSM** | **DoIT** | **GSM-Hard** | |
| LLAMA3-3B (Grattafiori et al., 2024) | | | | |
| FedIT | 35.42 | 41.48 | 48.20 | 41.70 (-11.86) |
| SMARTFED | **46.89** | **53.57** | **60.21** | **53.56** |
| LLAMA3-8B (Grattafiori et al., 2024) | | | | |
| FedIT | 47.55 | 56.14 | 64.92 | 56.20 (-13.82) |
| SMARTFED | **61.28** | **70.33** | **78.44** | **70.02** |
| QWEN3-8B (Yang et al., 2025) | | | | |
| FedIT | 50.25 | 58.54 | 66.49 | 58.43 (-13.82) |
| SMARTFED | **64.21** | **72.99** | **79.56** | **72.25** |

nese+Math task. As shown in Figure 10, MoLE assigns identical weights to all rank-wise components (0.36 for Chinese LoRA and 0.64 for Math LoRA). In contrast, MoRE allocates weights independently to them, enabling more granular exploitation of task-relevant knowledge. This fine-grained control facilitates precise knowledge integration, which translates into superior performance.

**Model Generalization Analysis.** To further examine model generalization, we evaluate SMARTFED on several recent, state-of-the-art backbones, including LLaMA3-

*Table 3.* Ablation study of SMARTFED.

| Configuration | Evaluation Task | | | Average |
|---|---|---|---|---|
| | MGSM | DoIT | GSM-Hard | |
| LLaMA2-7B (Touvron et al., 2023) | | | | |
| SMARTFED | **35.48** | **52.59** | **64.19** | **50.75** |
| w/o MoRE | 32.52 | 48.81 | 58.22 | 46.52 (-4.23) |
| w/o EEQA | 33.60 | 49.59 | 58.70 | 47.30 (-3.45) |
| LLaMA2-13B (Touvron et al., 2023) | | | | |
| SMARTFED | **53.18** | **63.01** | **72.65** | **62.95** |
| w/o MoRE | 48.32 | 58.71 | 66.95 | 58.00 (-4.95) |
| w/o EEQA | 50.25 | 60.20 | 68.46 | 59.64 (-3.31) |

3B (Grattafiori et al., 2024), LLaMA3-8B (Grattafiori et al., 2024), and Qwen3-8B (Yang et al., 2025). As shown in Table 2, SMARTFED consistently outperforms FedIT across all evaluation tasks. On LLaMA3-3B, SMARTFED improves the average performance from 41.70% to 53.56%, yielding an 11.86% gain. The advantage becomes even more pronounced on larger and newer backbones: SMARTFED achieves average gains of 13.82% on both LLaMA3-8B and Qwen3-8B. These improvements indicate that the benefits of SMARTFED are not tied to a specific architecture or model scale. Instead, by reusing task-relevant LoRA knowledge through fine-grained rank-wise routing, SMARTFED provides a general and transferable mechanism for resource-efficient federated fine-tuning.

**5.6. Ablation Study**

Finally, we conduct extensive ablation studies to assess the effectiveness of the two key techniques in SMARTFED: MoRE and EEQA. Table 3 shows that both techniques make substantial contributions to model performance. On LLaMA2-7B, removing MoRE and EEQA leads to average performance drops of 4.23% and 3.45%, respectively, with similar degradation observed on LLaMA2-13B (4.95% and 3.31%). These results underscore that MoRE effectively leverages fine-grained knowledge from rank-wise experts, while EEQA adaptively prioritizes more influential experts. This synergy consistently enhances overall performance.

## 6. Related Work

**LoRA-Based Federated Fine-Tuning.** Recent studies have explored integrating LoRA into federated fine-tuning frameworks. Pioneering works, such as FedIT (Zhang et al., 2024), combined FedAvg with LoRA for instruction tuning. Subsequent studies, including EcoLoRA (Liu et al., 2025a) and FedTT (Ghiasvand et al., 2025), have been introduced to enhance communication efficiency. To further address system heterogeneity, methods like FLoRA (Wang et al., 2024c), HETLoRA (Cho et al., 2024), FlexLoRA (Bai et al.,

2024), and HierFedLoRA (Liu et al., 2025b) propose diverse mechanisms, ranging from dynamic rank allocation to selective layer tuning, to accommodate devices with varying resource constraints. Additionally, DevFT (Wu et al., 2026a) proposed a developmental fine-tuning paradigm that progressively cultivates a powerful LLM from a compact base model to minimize resource overhead. However, despite their promise, these methods still operate under a "train-from-scratch" paradigm, which continues to impose substantial training overhead for edge devices.

**LoRA Knowledge Reuse.** To circumvent expensive retraining, several centralized approaches attempt to reuse knowledge from existing, task-adapted LoRA modules. LoRAHub (Huang et al., 2024a) learns scalar weights for linearly combining multiple LoRA modules, aiming for multi-task capabilities. Similarly, MoLE (Huang et al., 2024b) and LoRA-Flow (Wang et al., 2024a) utilize a router to dynamically integrate knowledge from different LoRA experts. However, these methods overlook data privacy concerns and, by treating LoRA modules as coarse-grained atomic units, fail to effectively exploit fine-grained subspace knowledge. Departing from prior designs, SMARTFED decomposes LoRA modules into rank-wise experts, enabling fine-grained knowledge reuse and precise adaptation while strictly preserving data privacy.

## 7. Conclusion

In this paper, we propose SMARTFED, a resource-efficient federated fine-tuning framework that reuses existing LoRA knowledge for LLM adaptation. Central to SMARTFED is the Mixture of Rank-Wise Experts, which decomposes LoRA into lightweight experts and activates them via a sparse, input-conditioned router. An Elastic Expert Quota Allocation strategy further improves knowledge utilization by adaptively assigning expert budgets across parameter matrices according to their task-specific importance. Extensive experiments demonstrate that SMARTFED significantly outperforms existing methods by a large margin, while substantially reducing the resource overhead on edge devices.

## Acknowledgements

This work is supported in part by the Science and Technology Development Fund of Macau (0107/2024/RIA2 and 0061/2025/RIB2), Joint Science and Technology Research Project with Hong Kong and Macau in Key Areas of Nansha District's Science and Technology Plan (EF2024-00180-IOTSC), and the Multi-Year Research Grant of University of Macau (MYRG-GRG2023-00211-IOTSC-UMDF and MYRG-GRG2024-00180-IOTSC). Zhijiang Guo is supported by the Guangdong Provincial Key Lab of Integrated Communication, Sensing and Computation for Ubiquitous Internet of Things (No. 2023B1212010007).

## Impact Statement

This work aims to make federated fine-tuning of LLMs more practical for resource-constrained edge devices by reducing training, communication, and energy costs while keeping data decentralized. These improvements may broaden access to task-specific LLM adaptation in privacy-sensitive or data-scarce settings, such as mobile, enterprise, and on-device applications. More broadly, we do not anticipate any additional risks beyond those already inherent in the use of LLMs. As with any LLM adaptation method, responsible data use, model evaluation, and deployment-specific privacy and security safeguards remain important.

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

# A. Theoretical Proofs

## A.1. Proof of Theorem 4.1

### 1. Noise Suppression Analysis

We analyze the expected squared norm of the accumulated task-irrelevant noise. First, for the dense baseline, using the assumptions that $\mathbb{E}[\xi_i^\top \xi_j] = 0$ for $i \neq j$ and $\mathbb{E}\|\xi_i\|^2 = \sigma^2$:

$$\mathbb{E}\|\Phi_{\text{dense}}\|^2 = \mathbb{E}\left\| \sum_{i \in \mathcal{S}_{\text{irr}}} \xi_i \right\|^2 = \sum_{i \in \mathcal{S}_{\text{irr}}} \mathbb{E}\|\xi_i\|^2 = |\mathcal{S}_{\text{irr}}|\sigma^2 = \rho M \sigma^2. \tag{14}$$

Next, for SMARTFED, the noise vector is $\Phi_{\text{SMARTFED}} = \sum_{i \in \mathcal{S}_{\text{irr}}} \hat{g}_i \xi_i$. Note that the gating weights $\hat{g}_i$ are determined by the routing strategy (or random selection), while $\xi_i$ represents intrinsic noise. Treating $\hat{g}_i$ as fixed conditional on the routing outcome, and assuming that the intrinsic noise is independent of the routing weights:

$$\mathbb{E}\left[\|\Phi_{\text{SMARTFED}}\|^2 \mid \{\hat{g}_i\}\right] = \sum_{i \in \mathcal{S}_{\text{irr}}} \hat{g}_i^2 \mathbb{E}\|\xi_i\|^2 = \sigma^2 \sum_{i \in \mathcal{S}_{\text{irr}}} \hat{g}_i^2. \tag{15}$$

Under the uninformative baseline, the routing rule is exchangeable with respect to expert indices. Conditioned on the total squared gating mass $S_2 = \sum_{j=1}^{M} \hat{g}_j^2$, the expected squared mass assigned to the irrelevant set is proportional to its size:

$$\mathbb{E}\left[ \sum_{i \in \mathcal{S}_{\text{irr}}} \hat{g}_i^2 \mid S_2 \right] = \frac{|\mathcal{S}_{\text{irr}}|}{M} S_2 = \rho S_2. \tag{16}$$

Substituting this back and applying the law of total expectation conditioned on $S_2$, we get:

$$\mathbb{E}\left[\|\Phi_{\text{SMARTFED}}\|^2 \mid S_2\right] = \sigma^2 \mathbb{E}\left[ \sum_{i \in \mathcal{S}_{\text{irr}}} \hat{g}_i^2 \mid S_2 \right] = \rho S_2 \sigma^2. \tag{17}$$

Comparing this to the dense baseline $\mathbb{E}\|\Phi_{\text{dense}}\|^2 = \rho M \sigma^2$, we obtain:

$$\mathbb{E}\left[\|\Phi_{\text{SMARTFED}}\|^2 \mid S_2\right] = \frac{\rho S_2 \sigma^2}{\rho M \sigma^2} \cdot \mathbb{E}\|\Phi_{\text{dense}}\|^2 = \frac{S_2}{M} \mathbb{E}\|\Phi_{\text{dense}}\|^2. \tag{18}$$

Taking expectation over the routing randomness further gives:

$$\mathbb{E}\left[\|\Phi_{\text{SMARTFED}}\|^2\right] = \rho \sigma^2 \mathbb{E}[S_2]. \tag{19}$$

This completes the first part of the proof.

### 2. Algebraic Cross-Term Elimination

Linear arithmetic (parameter merging) separately merges the LoRA factors. When applying the merged LoRA module to an input $\mathbf{x}$, the incremental update becomes:

$$\Delta W_{\text{Linear}} \mathbf{x} = \left( \sum_{n=1}^{N} \lambda_n \mathbf{B}_n \right) \left( \sum_{n'=1}^{N} \lambda_{n'} \mathbf{A}_{n'} \right) \mathbf{x} = \sum_{n=1}^{N} \lambda_n^2 \mathbf{B}_n \mathbf{A}_n \mathbf{x} + \sum_{n \neq n'} \lambda_n \lambda_{n'} \mathbf{B}_n \mathbf{A}_{n'} \mathbf{x}. \tag{20}$$

The second term represents $O(N^2)$ cross-task cross-terms (e.g., combining the projection-down matrix $\mathbf{A}$ from Task $n'$ with the projection-up matrix $\mathbf{B}$ from Task $n$), which are algebraically incorrect for the target manifold.

In contrast, SMARTFED treats each rank-one component as an atomic expert $E_i(\mathbf{x}) = (\mathbf{B}_{:,i} \mathbf{A}_{i,:})\mathbf{x}$. The aggregation is:

$$\Delta W_{\text{SMARTFED}} \mathbf{x} = \sum_{i=1}^{M} \hat{g}_i \cdot (\mathbf{B}_{:,i} \mathbf{A}_{i,:})\mathbf{x}. \tag{21}$$

Since the multiplication $\mathbf{B}_{:,i} \mathbf{A}_{i,:}$ occurs *before* summation, no cross-products of the form $\mathbf{B}_{:,i} \mathbf{A}_{j,:}$ ($i \neq j$) are ever generated. $\qquad \square$

### A.2. Proof of Proposition 4.2

Let $q_j^\star = \min(\alpha_j B, M)$ be the target continuous allocation, and assume $\sum_{j=1}^{J} \alpha_j = 1$ (as produced by our normalization). Algorithm 1 operates in two phases:

- **Phase 1 (flooring):** Assign $q_j^{(0)} = \lfloor q_j^\star \rfloor$. Since $0 \le q_j^\star \le M$, we have $q_j^{(0)} \in \{0, 1, \dots, M\}$ and

$$0 \le q_j^\star - q_j^{(0)} < 1. \tag{22}$$

- **Phase 2 (residual distribution):** Let the residual budget be $R = B - \sum_{j=1}^{J} q_j^{(0)}$. Since $\sum_{j=1}^{J} q_j^\star \le \sum_{j=1}^{J} \alpha_j B = B$ and $q_j^{(0)} \le q_j^\star$, we have $R \ge 0$. The algorithm increments a subset of coordinates that are not saturated (i.e., $\tilde{q}_j \le M$ always), yielding $\tilde{q}_j \ge q_j^{(0)}$ for all $j$ and $\sum_{j=1}^{J} \tilde{q}_j = \sum_{j=1}^{J} q_j^{(0)} + R \le B$.

We now bound the utility gap. Because $f$ is 1-Lipschitz on $[0, M]$, for each $j$,

$$f(q_j^\star) - f(q_j^{(0)}) \le |q_j^\star - q_j^{(0)}| < 1, \tag{23}$$

where we used (22). Multiplying by $\alpha_j$ and summing over $j$ gives

$$\sum_{j=1}^{J} \alpha_j \left( f(q_j^\star) - f(q_j^{(0)}) \right) \le \sum_{j=1}^{J} \alpha_j \cdot 1 = 1, \tag{24}$$

and hence

$$\sum_{j=1}^{J} \alpha_j f(q_j^{(0)}) \ge \sum_{j=1}^{J} \alpha_j f(q_j^\star) - 1. \tag{25}$$

Finally, Phase 2 only increases quota ($\tilde{q}_j \ge q_j^{(0)}$) while respecting the cap, and $f$ is non-decreasing on $[0, M]$, so $f(\tilde{q}_j) \ge f(q_j^{(0)})$ for all $j$. Therefore,

$$\sum_{j=1}^{J} \alpha_j f(\tilde{q}_j) \ge \sum_{j=1}^{J} \alpha_j f(q_j^{(0)}) \overset{(25)}{\ge} \sum_{j=1}^{J} \alpha_j f(q_j^\star) - 1. \tag{26}$$

This completes the proof. $\qquad\square$

## B. Acquisition of Skill-Specific LoRA Modules

In this section, we detail the task-specific LoRA modules employed in our experiments. For different skill-composition tasks, we utilize different types of LoRA modules:

- Chinese+Math task: Chinese Chat LoRA and English Math LoRA;

- Chinese+Code task: Chinese Chat LoRA and English Code LoRA;

- Math+Code task: English Math LoRA and English Code LoRA.

The acquisition process for these LoRA modules is as follows:

- **Chinese Chat LoRA:** We utilize the Okapi dataset (Lai et al., 2023), comprising 52K training examples, to train a LoRA module capable of understanding and generating Chinese text.

- **English Math LoRA:** This LoRA module is trained on MetaMathQA (Yu et al., 2024), an English-language dataset comprising 395K mathematical reasoning problems.

*Table 4.* Overview of skill-composition tasks, corresponding datasets, and evaluation metrics.

| Task | Skills Composed | Training Dataset | Testing Dataset | Evaluation Metric |
|---|---|---|---|---|
| Chinese Mathematical Reasoning | {Chinese, Math} | Math23K | MGSM | Accuracy |
| Chinese Code Generation | {Chinese, Code} | DoIT | DoIT | Pass@1 |
| Hard Math-Word Problems | {Math, Code} | MathCodeInstruct | GSM-Hard | Execution Accuracy |

- **English Code LoRA:** This LoRA module is fine-tuned on the Magicoder dataset (Wei et al., 2024), comprising 186K English-language code generation problems.

We integrate LoRA into the `Query` and `Value` matrices within attention modules. The LoRA rank $r$ is set to 32 and the scaling factor $\alpha$ is set to 64. We use the cosine warmup schedule and the peak learning rate is 1e-4. Each LoRA module is trained for 3 epochs with a warmup ratio of 0.04.

## C. More Implementation Details

**Federated Setup and Data Partitioning.** Following existing work (Ye et al., 2024; Tian et al., 2026), we establish a client pool comprising 20 devices. It is important to note that unlike traditional federated tasks where data heterogeneity is typically modeled by label distribution skew, instruction tuning involves generative tasks that lack explicit class labels. Consequently, defining non-IID distributions based on labels is inapplicable in this context. Therefore, consistent with established protocols in federated LLM fine-tuning such as FLoRA (Wang et al., 2024c) and OpenFedLLM (Ye et al., 2024), we adopt a random partitioning strategy to distribute the data. This setup ensures alignment with current community standards and allows for a focused evaluation of the knowledge composition mechanism itself.

Specifically, for the Chinese mathematical reasoning task (Chinese+Math), we partition the Math23K dataset (Wang et al., 2017) across these devices. The data is distributed uniformly at random, allocating approximately 1K samples to each device. For the Chinese code generation task (Chinese+Code), we partition the DoIT dataset (Song et al., 2025) across the devices. To ensure a fair comparison, we expand DoIT (Song et al., 2025) to 20K samples through web crawling, followed by manual verification using GPT-4 (Achiam et al., 2023), providing sufficient training data for participating devices. This expanded dataset is then evenly distributed, resulting in approximately 1K samples per device. Similarly, for experiments on hard math–word problems (Math+Code), we partition the MathCodeInstruct dataset (Wang et al., 2024b) among the devices, with each device receiving approximately 1K samples.

**Training Protocol.** In each training round, we randomly sample 10% of the devices to participate in the federated fine-tuning process, with each device performing 10 local update steps (Wu et al., 2026a; Ye et al., 2024). The training process continues for 20 rounds. The batch size is set to 16, the learning rate is 5e-4, and the optimizer used is AdamW. For our method, $K$ is set to 32. For the Chinese mathematical reasoning task, we use MGSM (Shi et al., 2023) as the testing set. For the Chinese code generation task, we use DoIT (Song et al., 2025) as the testing set. For the hard math-word problems, we use GSM-Hard (Gao et al., 2023) as the testing set. We report the accuracy on MGSM, pass@1 on DoIT, and execution accuracy on GSM-Hard. Table 4 summarizes the tasks, datasets, and evaluation metrics. To ascertain statistical significance and robustness, all reported results are averaged over three independent runs with different random seeds.

**Baseline Implementation and Adaptation.** To ensure reproducibility and fair comparison, we adopted the official open-source implementations for all baselines. For knowledge-reuse methods originally designed for centralized training, such as MoLE (Huang et al., 2024b) and LoRAHub (Huang et al., 2024a), we explicitly adapted them to the federated setting. Specifically, the trainable components inherent to these methods (e.g., the router network in MoLE and the composition weights in LoRAHub) are treated as local model parameters. During the federated process, these components are optimized locally on client data and then aggregated globally via FedAvg, ensuring they are synchronized. For consistency, all baselines utilized the same hyperparameters (e.g., participation rate, local update steps) as SMARTFED unless otherwise specified in their original papers. The federated fine-tuning process for baselines continues until model convergence is achieved.

# D. Baseline Description

We compare SMARTFED against two categories of baselines: Knowledge-Free methods (training from scratch) and Knowledge-Reuse methods (adapting existing LoRA modules). Detailed descriptions and our adaptation strategies for the federated setting are provided below.

## D.1. Knowledge-Free Methods

These methods train LoRA modules from scratch on the client's local data.

- **FedIT** (Zhang et al., 2024): A standard federated instruction tuning framework. It employs FedAvg to aggregate LoRA updates from clients, serving as a representative baseline for general-purpose federated LLM fine-tuning.

- **FwdLLM** (Xu et al., 2024b): A backpropagation-free federated fine-tuning method that utilizes zeroth-order optimization for gradient estimation, thereby significantly reducing the memory footprint on edge devices during training.

- **DoFIT** (Xu et al., 2024a): A domain-aware federated instruction tuning method. It is designed to alleviate catastrophic forgetting and handle data heterogeneity by incorporating domain-specific adaptation strategies.

- **FedAdapter** (Cai et al., 2023): An efficiency-focused method that optimizes the configuration of adapters to accelerate model convergence and reduce communication overhead.

- **AdaLoRA** (Zhang et al., 2023b): This is an adaptive budget allocation method designed for centralized training. We adapted it to the federated scenario by enabling clients to compute importance scores for LoRA ranks locally. The server then aggregates these scores to dynamically allocate the parameter budget (i.e., ranks) across parameter matrices.

## D.2. Knowledge-Reuse Methods

Since no existing federated methods explicitly reuse heterogeneous LoRA modules for LLM adaptation, we adapted several state-of-the-art centralized knowledge composition methods to the federated setting. For all methods below, we employ FedAvg to aggregate the learnable components (e.g., routers, composition weights) across clients.

- **Linear Arithmetic** (Zhang et al., 2023a): Linear Arithmetic is a training-free weight-space merging method that composes multiple pre-trained LoRA modules through simple arithmetic operations. Given a set of retrieved LoRA modules, it directly forms a merged adapter by taking a static weighted sum of their parameters, without introducing any input-dependent routing or additional fusion network. *Federated Adaptation:* In our federated setup, we freeze the backbone model and all candidate LoRA modules, and construct a single global merged LoRA using fixed/static coefficients over the retrieved modules. The same merged adapter is then shared across all clients and applied uniformly to all inputs, layers, and generation steps. This baseline therefore represents a simple static weight-space integration strategy without learning client-specific or input-conditioned expert selection.

- **LoRAHub** (Huang et al., 2024a): LoRAHub is a task-level LoRA composition method that reuses multiple pre-trained LoRA modules by learning a set of scalar coefficients. Given a target task, it composes candidate LoRA modules into a single integrated adapter and keeps the resulting composition weights fixed for all inputs and tokens during inference. *Federated Adaptation:* We freeze the backbone model and all candidate LoRA modules, and only treat the task-level composition vector $\mathbf{w} \in \mathbb{R}^N$ as the learnable component. In each communication round, clients locally optimize the composition weights on their private instruction data, and the server aggregates the resulting weight vectors via FedAvg. This yields a global static LoRA composition policy shared across clients without updating the LoRA modules.

- **MoLE (Mixture of LoRA Experts)** (Huang et al., 2024b): MoLE composes multiple pre-trained LoRA modules through hierarchical expert weighting. Specifically, it treats the LoRA modules at each layer as experts and learns a layer-wise gating function to assign composition weights, aiming to preserve the characteristics of individual LoRAs while enabling flexible expert composition. *Federated Adaptation:* We freeze the backbone model and all LoRA experts, and only optimize the layer-wise gating functions. Each client updates the MoLE gates on its local data, and the server aggregates the gate parameters via FedAvg. This learns a global layer-wise expert composition policy without sharing local data or updating the LoRA experts.

- **LoRA-Flow** (Wang et al., 2024a): LoRA-Flow performs dynamic LoRA fusion for generative tasks by producing fusion weights that vary across decoding steps and model layers. Different from static task-level fusion, its fusion gate conditions on the current prefix representation to determine how much each LoRA module should contribute at each generation step. *Federated Adaptation:* We freeze the model and all LoRA modules, and only treat the prefix-conditioned fusion gates as trainable parameters. Clients locally optimize these gates to learn token-/step-dependent fusion behaviors, and the server aggregates the gate updates via FedAvg, yielding a global dynamic fusion policy.

    Although both MoLE and LoRA-Flow learn lightweight gates over frozen LoRA modules, they differ in their fusion granularity and motivation: MoLE focuses on hierarchical layer-wise composition of trained LoRA experts, whereas LoRA-Flow focuses on prefix-conditioned, generation-step-dependent fusion for autoregressive generation.

## E. More Discussion

**Deployment Feasibility.** SMARTFED addresses key practical challenges in federated deployment by fundamentally reducing network overhead. Unlike traditional methods that require constant synchronization of massive model weights, our framework synchronizes only the tiny router parameters. Because the task-specific LoRA experts remain frozen and cached locally, the need for high-bandwidth synchronization is entirely eliminated. Consequently, this inherently small communication payload largely alleviates the need for aggressive compression protocols. Nevertheless, standard gradient compression techniques (e.g., quantization, sparsification) can still be seamlessly applied to the router updates to further minimize overhead in extreme edge environments.

**Robustness to Heterogeneity and System Failures.** While our primary evaluation follows standard benchmarks, SMARTFED exhibits inherent resilience to the practical challenges of federated deployment, owing to its decoupled architectural design:

- **Resilience to non-IID Data:** In traditional federated fine-tuning, data heterogeneity often leads to "client drift," where local updates diverge significantly from the global optimum. SMARTFED mitigates this by keeping the rank-wise experts *frozen*. The training is restricted to the lightweight router, which learns to map input semantics to expert combinations. Since the router learns a generalized semantic mapping rather than memorizing domain-specific features, it is less susceptible to overfitting local non-IID distributions compared to full-parameter or standard LoRA tuning. To empirically verify this property, we expand the DoIT dataset to 100 clients and simulate non-IID data via task-domain partitioning with a Dirichlet distribution parameterized by $\alpha = 0.5$. As shown in Table 5, SMARTFED consistently outperforms FedIT on LLaMA2-7B under both IID and non-IID settings. Notably, SMARTFED under non-IID data achieves 53.04%, which even surpasses FedIT under the IID setting (46.25%). This demonstrates that SMARTFED provides a higher performance ceiling and maintains strong utility despite highly skewed local data.

*Table 5.* Performance evaluation under IID and non-IID client distributions on LLaMA2-7B.

| Distribution | FedIT | SMARTFED |
|---|---|---|
| IID | 46.25 | **56.81** |
| non-IID | 44.72 | **53.04** |

- **Adaptability to Resource Divergence:** Our EEQA strategy inherently functions as a resource-aware adaptation mechanism. It enables each participating device to adaptively determine the optimal number of activated experts based on its local hardware constraints (e.g., memory and battery status). Specifically, high-resource devices can afford a larger expert budget to capture subtle semantic nuances, whereas resource-constrained devices can automatically prioritize and activate only the top-ranked, most critical experts identified by EEQA. This *importance-based prioritization* ensures that even devices with strict resource limits contribute highly valuable gradients from the most information-rich subspaces, thereby ensuring inclusive participation without stragglers bottlenecking the global system.

- **Scalability with Large-Scale Expert Pools.** SMARTFED keeps the expert-computation cost during inference primarily determined by the active quota $K$, since only $K$ rank-wise experts are executed after routing. The router scoring cost still scales with the size of the candidate expert pool $M$, but this overhead is lightweight compared with executing all LoRA experts and can be further reduced by preselection or approximate top-$K$ routing for very large pools.

*Table 6.* Robustness evaluation on GSM-Hard under noisy and mismatched LoRA retrieval.

| Retrieved LoRA | LLaMA2-7B | LLaMA2-13B |
|---|---|---|
| Math+Code (Ideal) | 64.19 | 72.65 |
| Math+Code+Chinese (Noise) | **64.25** | **72.83** |
| Math+General (Mismatch) | 58.97 | 69.02 |
| FedIT | 56.17 | 63.87 |

- **Mitigation of Negative Transfer.** Existing coarse-grained methods (like MoLE) risk negative transfer when an entire retrieved module contains conflicting knowledge. SMARTFED minimizes this risk through **Rank-Wise Decomposition**. By breaking modules down into fine-grained subspaces, our method can surgically discard the specific rank-one components that cause conflict while retaining the beneficial ones. This granularity turns potential negative transfer into positive synergy, as the router learns to suppress specific interference terms (as discussed in Theorem 4.1) that would otherwise degrade performance in coarse aggregation schemes. We evaluate this robustness on GSM-Hard by injecting a noisy LoRA into the expert pool and by retrieving a weakly relevant general LoRA. As shown in Table 6, adding a noisy Chinese LoRA even slightly improves results, indicating that the router can suppress irrelevant components while exploiting useful subspaces. Even under mismatched retrieval with Math+General LoRAs, SMARTFED still outperforms FedIT on both models, confirming its robustness to imperfect retrieval.

- **Generalization to Out-of-Distribution Domains.** A strict constraint of SMARTFED is that the rank-wise experts remain frozen to ensure efficiency. One might ask whether this imposes a *performance ceiling* when the target domain is significantly disjoint from the source tasks of the retrieved experts. We argue that the combinatorial capacity of rank-wise decomposition mitigates this issue. Even if no single expert perfectly matches the target domain, the router can synthesize a new, effective hypothesis space by recombining orthogonal subspaces from diverse experts—akin to how a basis set can reconstruct complex signals. Furthermore, the trainable router itself acts as a lightweight adapter, providing the necessary plasticity to align these frozen bases with the target manifold. To empirically examine this capability, we evaluate SMARTFED on broader tasks using LLaMA2-7B by integrating a general LoRA trained on Alpaca-GPT4 (Peng et al., 2023). As shown in Table 7, SMARTFED consistently outperforms FedIT across MMLU (Hendrycks et al., 2021), BBH (Suzgun et al., 2023), and HumanEval (Chen et al., 2021), demonstrating strong generalization across diverse domains. For extreme domain shifts, SMARTFED can naturally serve as a high-quality initialization (warm-start) for a brief subsequent phase of LoRA fine-tuning, offering a flexible trade-off between pure efficiency and maximum adaptation.

*Table 7.* Task generalization evaluation on broader benchmarks using LLaMA2-7B.

| Method | MMLU | BBH | HumanEval |
|---|---|---|---|
| FedIT | 42.45 | 39.28 | 17.59 |
| SMARTFED | **51.84** | **46.61** | **21.28** |

## F. More Experimental Results

### F.1. Effect of Active Expert Quota

We first analyze the effect of the active expert quota $K$ on GSM-Hard. As shown in Table 8, increasing $K$ steadily improves accuracy on both LLaMA2-7B and LLaMA2-13B. This trend indicates that activating more rank-wise experts increases the expressiveness of the composed update, allowing SMARTFED to capture finer-grained task-relevant knowledge. The improvement from $K = 16$ to $K = 64$ further suggests that the expert pool contains complementary subspaces that can be effectively exploited when a larger activation budget is available.

### F.2. Quota Allocation across Parameter Matrices

In this section, we present comprehensive analyses of expert quota allocation patterns across different parameter matrices under various models and tasks. Figure 11 illustrates the expert quota distribution for LLaMA2-7B on Chinese+Code and Math+Code tasks, while Figure 12 and Figure 13 demonstrate the allocation patterns for LLaMA2-13B and Qwen2-7B across three tasks, respectively. Our analysis reveals distinct distribution patterns that vary significantly across parameter

*Table 8.* Effect of the active expert quota $K$ on GSM-Hard.

| $K$ | LLaMA2-7B | LLaMA2-13B |
|---|---|---|
| 16 | 58.42 | 68.04 |
| 32 | 64.19 | 72.65 |
| 64 | 68.92 | 77.51 |

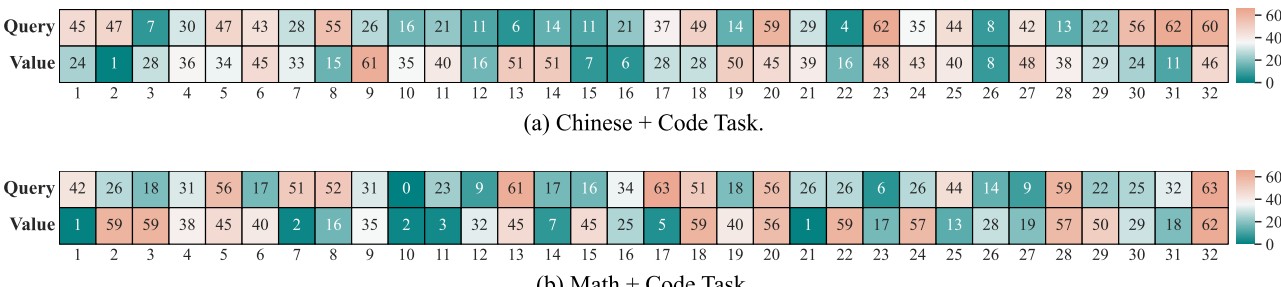

*Figure 11.* Distribution of expert quotas across different tasks in LLaMA2-7B.

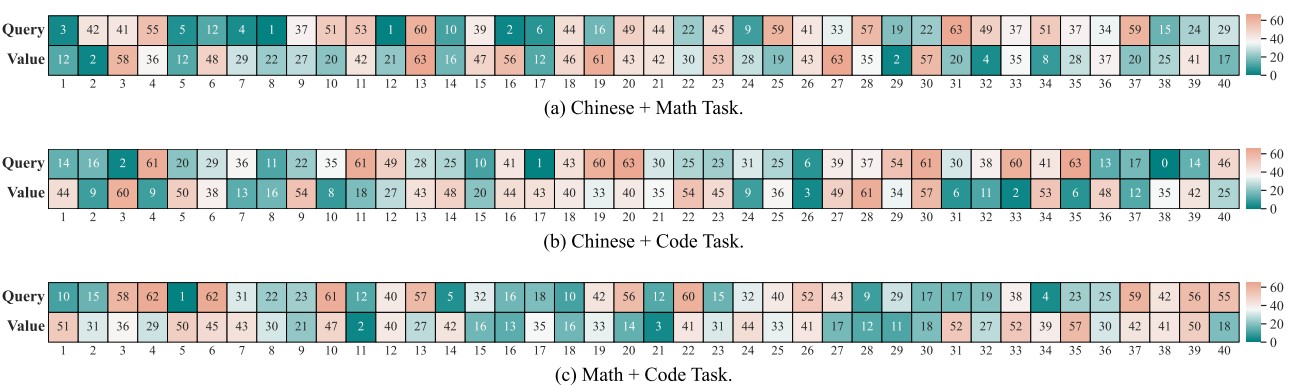

*Figure 12.* Distribution of expert quotas across different tasks in LLaMA2-13B.

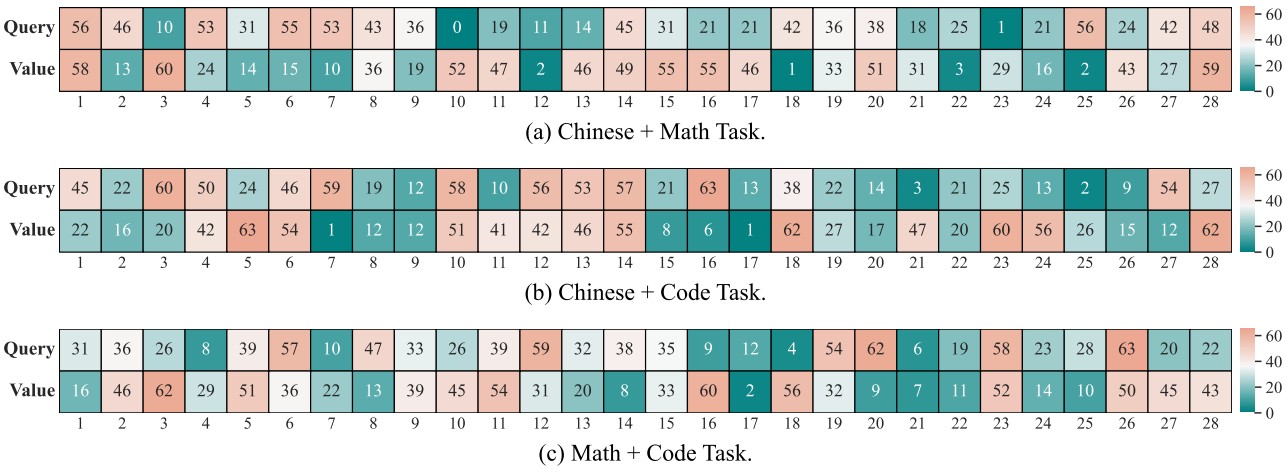

*Figure 13.* Distribution of expert quotas across different tasks in Qwen2-7B.

matrix types, layers, tasks, and model architectures. These heterogeneous allocation patterns validate the effectiveness of our proposed EEQA, which adaptively identifies crucial experts and consequently enhances knowledge utilization.

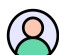 User Input

> Carl's favorite food is cheese. This week, he eats a sandwich for lunch every day, using 2 slices of cheese per sandwich. On 3 days of the week, he has cheese and omelet for breakfast, with each omelet using 1 more slice of cheese than each sandwich. He made a large portion of mac and cheese for several dinners this week, using 8 slices of cheese. **How many slices of cheese did he use in total?**

SmartFed Output:

> Carl eats sandwiches all 7 days of the week, using 2 slices of cheese per sandwich, so sandwiches used 7 * 2 = 14 slices of cheese in total. He had omelets on 3 days, with each omelet using 1 more slice than a sandwich, which means 2 + 1 = 3 slices, so omelets used 3 * 3 = 9 slices of cheese in total. The mac and cheese used 8 slices of cheese. In total, he used 14 + 9 + 8 = 31 slices of cheese this week. \n#### 31\n The answer is: 31. 👍

MoLE Output:

> Carl eats sandwiches 5 days a week, using 2 slices of cheese per sandwich, so sandwiches used 5 * 2 = 10 slices of cheese in total. He had omelets on 3 days, with each omelet using 1 more slice than a sandwich, which means 2 + 1 = 3 slices, so omelets used 3 * 3 = 9 slices of cheese in total. The mac and cheese used 8 slices of cheese. In total, he used 10 + 9 + 8 = 27 slices of cheese this week. \n#### 27\n The answer is: 27 ❌

*Figure 14.* Case study on the Chinese+Math task (LLaMA2-7B). Chinese query and responses are translated to English for presentation clarity.

### F.3. Case Study

To further demonstrate the effectiveness of SMARTFED, we present a case study on the Chinese+Math task using LLaMA2-7B (Figure 14). This example illustrates how fine-grained knowledge fusion enables more precise comprehension of user queries. Unlike MoLE, which treats each LoRA module as a whole and assigns uniform weights, SMARTFED employs rank-wise experts to selectively integrate knowledge from task-related subspaces. This enhanced granularity allows SMARTFED to capture subtle semantic nuances, resulting in precise solutions. In contrast, MoLE exhibits significant hallucinations—notably misinterpreting fundamental facts like

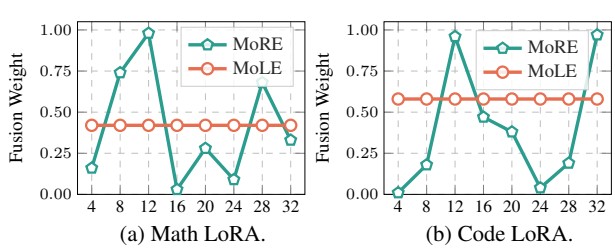

*Figure 15.* Average fusion weights of rank-wise experts in the Query matrix at the third layer for the Math+Code task (LLaMA2-7B).

the number of days in a week—highlighting the limitations of coarse-grained fusion approaches. Furthermore, Figure 16 presents an illustrative example from the Math+Code task, where Figure 15 demonstrates the average fusion weights of rank-wise components for this sample. We observe that MoLE fails to retain crucial user-provided information, resulting in inaccurate auxiliary program generation and consequently erroneous solutions. In contrast, SMARTFED, through a more flexible knowledge reuse strategy, exhibits enhanced comprehension of user requirements and robust information retention. This, in turn, facilitates the generation of precise programs and accurate solutions, demonstrating SMARTFED's capability to handle complex mathematical problems with high fidelity. Overall, these results underscore the advantages of SMARTFED in delivering more reliable and precise responses through fine-grained knowledge integration.

## G. Discussion of Concurrent and Recent Related Works

**Comparison with LoRACoE.** LoRACoE (Li et al., 2025) focuses on centralized training, updating LoRA alongside the router. In contrast, SMARTFED targets resource-constrained federated environments, strictly reusing frozen LoRA modules from disparate tasks and thereby circumventing the heavy retraining of LoRA. Moreover, unlike LoRACoE's static expert

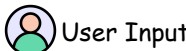 User Input

Martha is knitting winter wear for her 2189997 grandchildren. They're triplets, so they're all the same size. She wants to make a hat, scarf, sweater, mittens, and socks for each of them. It takes 2 skeins of wool to make a hat, 4 for a scarf, 12 for a sweater, 1 for a pair of mittens, and 2 for a pair of socks. **How many skeins of wool will she need to buy?**

SmartFed Output:

```
Code:"
def solution():
    \"\"\"Martha is knitting winter wear for her 2189997
grandchildren. They're triplets, so they're all the same
size. She wants to make a hat, scarf, sweater, mittens,
and socks for each of them. It takes 2 skeins of wool to
make a hat, 4 for a scarf, 12 for a sweater, 1 for a pair
of mittens, and 2 for a pair of socks. How many skeins
of wool will she need to buy?\"\"\"
    num_grandchildren = 2189997
    skeins_per_hat = 2
    skeins_per_scarf = 4
    skeins_per_sweater = 12
    skeins_per_mittens = 1
    skeins_per_socks = 2
    skeins_per_grandchild = skeins_per_hat +
skeins_per_scarf + skeins_per_sweater +
skeins_per_mittens + skeins_per_socks

    total_skeins = num_grandchildren *
skeins_per_grandchild

    result = total_skeins
    return result
"
```

**Answer: 45989937.0**

MoLE Output:

```
Code:"
def solution():
    \"\"\"Martha is knitting winter wear for her 2189997
grandchildren. They're triplets, so they're all the same
size. She wants to make a hat, scarf, sweater, mittens,
and socks for each of them. It takes 2 skeins of wool to
make a hat, 4 for a scarf, 12 for a sweater, 1 for a pair
of mittens, and 2 for a pair of socks. How many skeins
of wool will she need to buy?\"\"\"
    num_grandchildren = 2189997
    skeins_per_hat = 2
    skeins_per_scarf = 4
    skeins_per_sweater = 12
    skeins_per_mittens = 1

skeins_per_grandchild = skeins_per_hat +
skeins_per_scarf + skeins_per_sweater +
skeins_per_mittens

    total_skeins = num_grandchildren *
skeins_per_grandchild

    result = total_skeins
    return result
"
```

**Answer:41609943.0**

*Figure 16.* Case study on the Math+Code task (LLaMA2-7B).

activation, SMARTFED introduces EEQA, which dynamically distributes expert budgets across parameter matrices.

**Comparison with Private Routing.** Methods such as SEQR (Fleshman & Van Durme, 2025) perform unsupervised routing to select an entire LoRA module. Conversely, SMARTFED uses federated learning to collaboratively train a lightweight router, enabling fine-grained, rank-level composition that actively mitigates cross-task interference.

## H. The Use of Large Language Models

During manuscript preparation, a large language model was occasionally employed as an auxiliary assistant to refine language expression, such as improving sentence fluency and enhancing readability. The model was not involved in generating original research contributions: it did not participate in formulating research questions, designing methodologies, conducting experiments, analyzing results, or drafting substantive scientific content. All core intellectual work, including the development of ideas, execution of experiments, and interpretation of findings, was carried out independently by the authors. Any linguistic suggestions offered by the LLM were critically reviewed and selectively incorporated, ensuring that accuracy, originality, and scholarly integrity were fully maintained. The authors alone bear responsibility for the research content and conclusions, and the LLM is not listed as a contributor or author.

