# OpenReview forum: "Don't Reinvent the Wheel, Just Realign the Spokes: Resource-Efficient Federated Fine-Tuning via Rank-Wise Expert Assembly"
_ICML.cc/2026/Conference — ICML 2026 spotlight_

### Official Review · Reviewer_T8ZG · 2026-02-20

**Soundness:** 4
**Presentation:** 4
**Significance:** 4
**Originality:** 3
**Overall Recommendation:** 6
**Confidence:** 4

**Summary:**

This paper introduces the SmartFed framework, which circumvents federated training of parameters directly and instead composes a Mixture of Rank-Wise Experts (MoRE) using existing LoRAs. The method learns to budget and route between the experts using the proposed Elastic Expert Quota Allocation (EEQA). The efficient framework addresses two limitations of existing approaches: 1) coarse adaptation and 2) scalability, which are respectively addressed by MoRE and EEQA. An importance score is used to measure the impact of individual experts across the model, driving the allocation of compute to areas with larger impacts on performance. The experiments demonstrate superior performance over several baselines across three models.

**Compliance With Llm Reviewing Policy:**

Affirmed.

**Final Justification:**

This paper is a clear accept. The original submission was already strong, and the authors were able to fully address my concerns during the discussion period. I therefore increased my score from 5 to 6.

**Key Questions For Authors:**

1) Why not include MoRE in Figure 3b?
2) Was the budget chosen so that the average activated rank (Table 1) would be 32? Seeing a performance curve with more or less budget would be informative.
3) Could the importance scores be used to prune experts completely, such that memory use is reduced?

**Limitations:**

yes

**Strengths And Weaknesses:**

**Strengths**

Soundness:

1) The presented claims are well supported by both theoretical and experimental results. The authors used a wide range of baselines and three LMs for their experiments.

2) The experimental design included performance comparisons, efficiencies, and ablations, directly testing the claims.

Presentation:

3) The paper is well written and easy to follow.

4) The figures and tables are visually appealing and clear.

Significance:

5) The experiments support that SmartFed does improve performance when compared to commonly used alternatives.

6) Deriving rank-wise components from **existing** LoRAs is unique as far as I know, and could introduce many new angles in LoRA composition.

7) The insights into the heterogeneity in LoRA importance across model layers are very interesting and could influence future research.

Originality:

8) While there is existing research using rank-wise LoRA components or budget constraints, SmartFed does introduce a novel combination of these techniques, especially by enabling pre-existing LoRAs to be used within the framework.

**Weaknesses**

Soundness:

1) Not a major issue, but the three LMs were all of similar size (7-13B). I would expect the benefits of the approach to scale with model size, and it would be interesting to see if the sparse selection of rank-wise components would work for smaller models (1-3B) where LoRAs might have higher importance.

Presentation:

2) Some more related works could better frame the proposed framework:
- Rank-wise LoRA components ([LoRACoE](https://aclanthology.org/2025.emnlp-main.1594/))
- Routing using existing LoRAs in a private manner (discusses multiple methods including Arrow, SpectR, and SEQR) ([SEQR](https://arxiv.org/abs/2509.18093))

Significance:

N/A

Originality:

N/A

---

> ### Author Rebuttal · Authors · 2026-03-29
>
> We sincerely thank the reviewer for the positive and constructive feedback. Below are our point-by-point responses to your concerns.
>
> > **W1: Effectiveness on Smaller Models (1-3B).**
>
> Thanks for the constructive comment. We evaluate SmartFed on LLaMA3-3B (Table A), where it consistently outperforms FedIT, achieving an average performance gain of 11.86%. This improvement stems from:
>
> - **Noise Suppression:** Smaller models lack the inherent capacity to absorb task-irrelevant noise. SmartFed operates at the rank-wise level. This eliminates cross-task interference, preventing the limited representational capacity from being overwhelmed by noise.
> - **Resource Prioritization:** EEQA dynamically allocates the limited expert budget to the most critical components. This ensures the model's restricted capacity is utilized with maximum efficiency.
>
> **Table A: Performance Evaluation on LLaMA3-3B.**
>
> | Method   | MGSM      | DoIT      | GSM-Hard  | Average   |
> | -------- | --------- | --------- | --------- | --------- |
> | FedIT    | 35.42     | 41.48     | 48.20     | 41.70     |
> | SmartFed | **46.89** | **53.57** | **60.21** | **53.56** |
>
>
>
> > **W2: Inclusion of Recent Related Works.**
>
> We thank the reviewer for highlighting these highly relevant concurrent works. We will include the discussion below in our related work section.
>
> **1. Comparison with LoRACoE:**
>
> - **Federated vs. Centralized:** LoRACoE focuses on centralized training, updating LoRA alongside the router. In contrast, SmartFed targets resource-constrained federated environments. We strictly reuse frozen LoRA modules from disparate tasks, circumventing the heavy retraining of LoRA.
> - **Dynamic vs. Static Allocation:** Unlike LoRACoE’s static expert activation, SmartFed introduces EEQA, which dynamically distributes expert budgets across matrices.
>
> **2. Comparison with Private Routing (SEQR, Arrow, SpectR):**
>
> - **Fine-Grained Composition vs. Coarse Selection:** Methods like SEQR perform unsupervised routing to select an entire LoRA module. Conversely, SmartFed uses FL to collaboratively train a lightweight router. This enables fine-grained, rank-level composition, actively mitigating cross-task interference.
>
>
>
> > **Q1: Include MoRE in Figure 3b.**
>
> We thank the reviewer for this valuable suggestion. Figure 3b originally aimed to highlight a core baseline dilemma: Linear Arithmetic is efficient but suffers from severe cross-task interference, whereas MoLE mitigates this noise but incurs exploding computational costs. As requested, we evaluate MoRE's inference latency in Table B. By utilizing a sparse rank-wise selection mechanism, MoRE maintains a highly stable, near-constant latency. This confirms that MoRE preserves the expressiveness of dynamic routing while guaranteeing scalability.
>
> **Table B: Inference Latency Comparison on LLaMA2-7B.**
>
> | Method            | N=2   | N=4   | N=8   | N=16   |
> | ----------------- | ----- | ----- | ----- | ------ |
> | Linear Arithmetic | 45.95 | 45.95 | 45.95 | 45.95  |
> | MoLE              | 49.52 | 54.90 | 84.71 | 112.64 |
> | **MoRE**          | 50.31 | 51.24 | 53.87 | 54.65  |
>
>
>
> > **Q2: Impact of Active Expert Budget.**
>
> The reviewer correctly observes that in Table 1, we strictly constrained the average activated rank per matrix to 32. This was a deliberate design choice to ensure a fair comparison. Following your suggestion, we evaluate SmartFed with varying expert budgets. Table C shows that performance scales positively with K. Activating more experts expands the hypothesis space, enhancing the expressiveness of the composed update to capture finer-grained, task-relevant knowledge.
>
> **Table C: Impact of the Number of Active Experts on GSM-Hard.**
>
> | K    | LLaMA2-7B | LLaMA2-13B |
> | ---- | --------- | ---------- |
> | 16   | 58.42     | 68.04      |
> | 32   | 64.19     | 72.65      |
> | 64   | 68.92     | 77.51      |
>
>
>
> > **Q3: Feasibility of Permanent Expert Pruning.**
>
> We appreciate this insightful comment. While pruning reduces memory footprints, it introduces a fundamental tension between global importance and per-sample necessity. Since required experts vary highly across inputs, static pruning inadvertently discards critical specialist experts, collapsing the pool's fine-grained diversity. To verify this, we conduct a hard-pruning ablation. As shown in Table D, this static pruning causes a noticeable performance degradation compared to dynamic routing (No-Prune).
>
> **Table D: Impact of Expert Pruning on LLaMA2-7B.**
>
> | Setting  | MGSM      | DoIT      | GSM-Hard  |
> | -------- | --------- | --------- | --------- |
> | No-Prune | **35.48** | **52.59** | **64.19** |
> | Prune    | 34.72     | 51.45     | 62.18     |
>
> ---
>
> Finally, thanks again for your time and helpful comments. We hope your concerns are well addressed. For any other questions, we are happy to discuss them and make further clarifications.

---

> > ### Author Rebuttal · Reviewer_T8ZG · 2026-03-31
> >
> > The authors have addressed all of my concerns in detail. These include:
> >
> > - Adding a Llama-3B evaluation (weakness 1)
> > - Expanding related works (weakness 2)
> > - Included MoRE in fig 3 (Q1)
> > - Evaluated approach with different budgets (Q2)
> > - Measured impact of pruning (Q2)
> >
> > Together, these additions will make the paper even stronger, and I have raised my scores accordingly.

---

> > > ### Author Response · Authors · 2026-04-01
> > >
> > > Thank you for your thorough review and for raising your score! We are thrilled that the new evaluations and analyses have fully addressed your concerns. Your constructive feedback has genuinely made our paper much stronger, and we will ensure all these additions are included in the final version.

---

### Official Review · Reviewer_wUJk · 2026-03-07

**Soundness:** 3
**Presentation:** 4
**Significance:** 4
**Originality:** 3
**Overall Recommendation:** 6
**Confidence:** 3

**Summary:**

This paper studies federated LLM adaptation in a setting where many useful task-specific LoRA modules already exist. Current federated methods train new adapters from scratch. Thus, the authors proposed SmartFed, which reuses retrieved LoRA modules and only trains a lightweight router on clients. Its two main technical components are: (1) MoRE, which decomposes each LoRA module into rank-wise experts and performs sparse input-conditioned routing over them, and (2) EEQA, which allocates expert budgets unevenly across parameter matrices based on an importance score. The paper presents a theoretical justification and experiments on three model families and three skill-composition tasks, showing consistent gains over both train-from-scratch federated baselines and adapted LoRA-reuse baselines, while incurring lower overhead, training time, and energy cost.

**Compliance With Llm Reviewing Policy:**

Affirmed.

**Final Justification:**

**My final recommendation is 6: Strong Accept.**

I find the paper technically sound, practically significant, and clearly presented. Its main strength is addressing an important and realistic problem in federated LLM fine-tuning: how to effectively reuse public LoRA modules without suffering from the noise of simple parameter arithmetic or the coarse granularity and cost of module-level routing.

The proposed shift from module-level experts to rank-wise experts is intuitive, technically meaningful, and well supported by both the ablations and the main empirical results. I also appreciate that SmartFed demonstrates strong gains over a broad set of baselines while offering substantial efficiency advantages in convergence speed, communication cost, and energy consumption. The paper is also generally well written and easy to follow.

At the same time, I initially had several concerns regarding scalability to more realistic federated settings, generalization to newer backbones and broader tasks, sensitivity to the quality of the retrieved LoRA pool, and the underdeveloped privacy discussion. The authors’ rebuttal was helpful and addressed these concerns sufficiently.

In particular, the added experiments on larger-scale and non-IID settings strengthened my confidence in the scalability claims; the new results on more recent models and broader benchmarks improved my view of the method’s generality. And the additional analysis on noisy or weakly relevant retrieved LoRAs made the robustness of the approach much more convincing. I also appreciate that the authors addressed the privacy issue with extra evidence, even though it was not one of my main requested questions.

**Overall, while some limitations remain, the rebuttal improved my assessment. I also considered the other reviewers’ feedback and the authors’ corresponding responses. Taken together, I believe its strengths outweigh the weaknesses. Hence, I raise my score accordingly.**

**Key Questions For Authors:**

**I would be willing to raise my score if the authors can (partly) address the following questions/concerns:**

- The current experiments use a pool of 20 clients with 10% participation per round, i.e., effectively only two active clients in each round. Could the authors provide additional experiments that SmartFed scales to more realistic federated settings, for example, with a larger client population and/or more variable participation rates?

- The experiments are conducted only on Llama-2-7B/13B and Qwen2-7B. Could the authors provide evidence that SmartFed also works well with more recent, larger open LLM backbones?

- How sensitive is SmartFed to the quality of the retrieved LoRA pool? In particular, what happens when the retrieved modules are only weakly relevant, highly redundant, substantially mismatched to the target task, or when no clearly relevant expert can be retrieved at all?

**Meanwhile, I might have misunderstood some aspects of the paper, so I would welcome the authors’ clarification where appropriate.**

**Limitations:**

Yes

**Strengths And Weaknesses:**

**Strengths:**

1. The paper addresses the problem: reusing public LoRA modules in federated fine-tuning is a sensible direction, and the paper makes a reasonable case that existing reuse strategies are either too noisy (linear arithmetic) or too coarse-grained and expensive (module-level routing).

2. The move from module-level experts to rank-wise experts is the main technical contribution, and it is both intuitive and reasonably well motivated. The ablations also support that both MoRE and EEQA matter: removing either component leads to clear performance drops.

3. The empirical results are good. SmartFed outperforms a broad set of baselines across multiple LLM sizes and types, including LLama2-7B/13B and Qwen2-7B. The reported efficiency gains are substantial: faster convergence, much lower communication, and lower energy use. The method also seems to be data-efficient, which is a meaningful practical advantage in FL settings.

4. The paper is generally well written. The motivation is clear, the method section is easy to follow, and the figures help explain the design choices.

**Weaknesses:**

1. The current setup uses a relatively small client pool with limited participation per round. The authors didn't test the harder settings that motivate FL in practice, such as stronger non-IID heterogeneity or larger-scale participation. As a result, the scalability and robustness claims are not yet fully convincing.

2. The experiments are conducted on Llama-2-7B/13B and Qwen2-7B, which are older-generation open LLM backbones. While this is sufficient for an initial validation, it is still unclear whether the same conclusions would hold on more recent and stronger open models.

3. The task setting is still a bit narrow. The experiments focus on three skill-composition tasks built around Chinese/Math/Code combinations. It remains unclear how well the method works when the retrieved LoRAs are only weakly related to the target task or when the target task is less naturally compositional.

4. The privacy discussion is underdeveloped. The paper emphasizes privacy preservation (e.g., in the abstract and Appendix E), partly because only router updates and scalar importance scores are communicated, but these claims are not directly evaluated or theoretically proven. Concrete results would make these claims much more convincing.

---

> ### Author Rebuttal · Authors · 2026-03-29
>
> We sincerely thank the reviewer for the positive and constructive feedback. Below are our point-by-point responses to your concerns.
>
>
>
> > **W1&Q1: Client Scale and Data Heterogeneity.**
>
> Thanks for the valuable comment. We evaluate scalability by expanding DoIT to 100 clients and simulate Non-IID through task-domain partitioning (α=0.5). SmartFed consistently outperforms FedIT on LLaMA2-7B (Table A). Notably, SmartFed's performance under Non-IID conditions (53.04) surpasses FedIT's IID performance (46.25). This demonstrates that SmartFed provides a higher performance ceiling, guaranteeing high utility despite highly skewed local data.
>
> **Table A: Scalability and Data Heterogeneity Analysis (LLaMA2-7B).**
>
> | Distribution | FedIT | SmartFed  |
> | ------------ | ----- | --------- |
> | IID          | 46.25 | **56.81** |
> | Non-IID      | 44.72 | **53.04** |
>
>
>
> > **W2&W3&Q2: Performance Across Recent Architectures and Broader Tasks.**
>
> Thanks for these constructive comments.
>
> - **Model Generalization:** We implement SmartFed on newer models (Table B).  SmartFed consistently outperforms FedIT on all models (e.g., 72.25 vs. 58.43 on Qwen3), demonstrating that SmartFed is a general framework for federated fine-tuning.
>
>   **Table B: Model Generalization Analysis.**
>
>   | Model     | Method   | MGSM      | DoIT      | GSM-Hard  | Average   |
>   | --------- | -------- | --------- | --------- | --------- | --------- |
>   | LLaMA3-8B | FedIT    | 47.55     | 56.14     | 64.92     | 56.20     |
>   |           | SmartFed | **61.28** | **70.33** | **78.44** | **70.02** |
>   | Qwen3-8B  | FedIT    | 50.25     | 58.54     | 66.49     | 58.43     |
>   |           | SmartFed | **64.21** | **72.99** | **79.56** | **72.25** |
>
> - **Task Generalization:** We extend our evaluation to broad benchmarks and add a general LoRA to the expert pool (Table C). SmartFed consistently outperforms FedIT. This proves that even lacking a matched expert, SmartFed dynamically recombines fine-grained rank-wise subspaces to achieve effective task adaptation.
>
>   **Table C: Task Generalization Analysis (LLaMA2-7B).**
>
>   | Method   | MMLU      | BBH       | HumanEval |
>   | -------- | --------- | --------- | --------- |
>   | FedIT    | 42.45     | 39.28     | 17.59     |
>   | SmartFed | **51.84** | **46.61** | **21.28** |
>
>
>
> > **W4: Privacy Discussion.**
>
> We appreciate the reviewer's suggestion. To verify compatibility with privacy-preserving algorithms, we evaluate SmartFed under Differential Privacy (DP) guarantees. SmartFed consistently outperforms FedIT, incurring a mere 1.89% performance drop compared to FedIT's 3.95% degradation. Because optimization is restricted to the lightweight router, SmartFed requires substantially less noise injection to satisfy the same privacy budget compared to standard fine-tuning methods.
>
> **Table D: Performance Evaluation on MGSM (LLaMA2-7B).**
>
> | Setting | FedIT | SmartFed  |
> | ------- | ----- | --------- |
> | No-DP   | 30.97 | **35.48** |
> | DP      | 27.02 | **33.59** |
>
>
>
> > **Q3: Analysis of the Retrieved LoRA Pool.**
>
> Thanks for the insightful comment.
>
> - **Mismatched/Redundant Modules:** Theorem 4.1 proves that SmartFed suppresses irrelevant noise. We validate this on GSM-Hard by injecting an irrelevant Chinese LoRA into the expert pool. As shown in Table E, the performance remains stable and even improves in certain cases, confirming its robustness.
>
>   **Table E: Robustness to Noisy Modules.**
>
>   | **Retrieved LoRA Pool**       | **LLaMA2-7B** | **LLaMA2-13B** | Qwen2-7B  |
>   | ----------------------------- | ------------- | -------------- | --------- |
>   | Math + Code                   | 64.19         | 72.65          | **63.62** |
>   | Math + Code + Chinese (Noise) | **64.25**     | **72.83**      | 63.55     |
>
> - **Weakly Relevant Modules:** We test SmartFed under a suboptimal candidate pool by replacing the Code LoRA with a general LoRA (Table F). SmartFed still achieves strong performance despite the absence of a well-matched expert, outperforming FedIT. This robustness comes from rank-wise decomposition, which enables SmartFed to recombine useful fine-grained subspaces and form a workable hypothesis space for the target task.
>
>   **Table F: Robustness to Weakly Relevant Modules (GSM-Hard).**
>
>   | Setting        | **LLaMA2-7B** | **LLaMA2-13B** |
>   | -------------- | ------------- | -------------- |
>   | Math + Code    | 64.19         | 72.65          |
>   | Math + General | 58.97         | 69.02          |
>   | FedIT          | 56.17         | 63.87          |
>
> - **LoRA Availability:** In practice, the open-source ecosystem already provides a large and rapidly growing pool of task-specific LoRAs. Moreover, SmartFed does not require exact task-matched modules—only reasonably related reusable ones.
>
>
>
> ---
>
> Finally, thanks again for your time and helpful comments. We hope your concerns are well addressed. For any other questions, we are happy to discuss them and make further clarifications.

---

> > ### Author Rebuttal · Reviewer_wUJk · 2026-03-31
> >
> > Thank you so much for providing thorough answers to my concerns and questions.
> >
> > The additional experiments on scalability and data heterogeneity are convincing and addressed my points.
> >
> > The new analysis of the retrieved LoRA pool also helps clarify the method’s sensitivity to retrieval quality.
> >
> > In addition, I appreciate that the authors also addressed the privacy weakness I raised, even though it was not listed among my key questions. The expanded privacy discussion is helpful, as it clarifies the scope of the paper’s privacy-preservation claims.
> >
> > I have also considered the questions raised by other reviewers and the authors’ corresponding responses, which further improved my overall confidence in the paper.
> >
> > Overall, I consider revising my score to a WOW one, as the new evidence meaningfully strengthens the paper’s significance and resolves all my main concerns, supporting a stronger case for acceptance.
> >
> > Cheers!

---

> > > ### Author Response · Authors · 2026-04-01
> > >
> > > This is incredibly encouraging news! We sincerely thank you for your highly positive feedback and for the extra time you took to read through the other reviews. We are thrilled that the new experiments and discussions have fully addressed your concerns. We deeply appreciate your strong support and your consideration for such a high score.
> > >
> > > Cheers!

---

### Official Review · Reviewer_NsPn · 2026-03-07

**Soundness:** 3
**Presentation:** 4
**Significance:** 3
**Originality:** 3
**Overall Recommendation:** 5
**Confidence:** 4

**Summary:**

Your paper introduces SmartFed, which is a framework for resource-efficient federated fine-tuning of LLMs. One of the key ideas is to simply reuse knowledge from pre-trained LoRA adapters and to dynamically compose them through a routing mechanism. Your method thus aims to avoid training new LoRAs from scratch.

**Compliance With Llm Reviewing Policy:**

Affirmed.

**Final Justification:**

**The authors have contributed a solid piece of work and I recommend clear acceptance**.

I have nothing to add, the rebuttal was solid as well. Other reviewers agree, too.

**Key Questions For Authors:**

Please see questions in Strengths and Weaknesses.

**Limitations:**

yes

**Strengths And Weaknesses:**

Thank you for your submission! Overall, I have understood the concept very well and have the following remarks:

### **1. Strengths**:
- **clear theory and concept**: the new idea is rank-wise decomposition of LoRA modules, which you present very clearly. Your method is conceptually interesting as it avoids destructive cross-task interactions in parameter merging. Your solution is quite elegant and addresses some issues in traditional task vector merging approaches.
- **strong efficiency improvements**: the results are one of the most convincing aspects of the paper. Faster convergence and much lower energy consumption makes this very appealing for edge compute cases.
- **principled ablation studies**: Since your method includes several steps, a principled ablation study is required to understand the contribution of each component. I think your ablations are detailed and make your method and results convincing.
- **many baselines**: You compare against several baselines, which you have adapted to the FL setting, and you show that your method outperforms them.

There are more results and insights (particularly regarding typical FL topics) in the appendix which seem to solidify your work.

### **2. Weaknesses**:

Soft weaknesses are *italic*, while strong weaknesses are **bold**:
- *novelty*: it seems to me that your concept is an adaptation/improvement of LoRA merging. Your method builds heavily on MoLE and LoRA-Flow, and introduces novelty mainly in decomposing LoRA modules into rank-wise components. The results however seem to justify this adaptation, hence this is just a minor weakness.
- *baseline adaptation fairness*: thanks for providing many baselines, however, several centralized methods are adapted for the FL case. Because of that, it is unclear whether these adaptations represent the strongest possible versions of the baselines. This should be discussed in the paper. Can you give some comments to this? Also, hyperparameters for the baselines are **not given**. If you perform such an adaptation, it would be required to give them for reproducibility.
- **use case unclear**: for me, it is a bit counter-intuitive to do "public LoRA merging" in FL. Is the whole point of FL not to do distributed learning with **private data**, i.e., data which is usually not found anywhere else? In that case, the argumentation about privacy falls flat as you basically combine "off-the-shelf" LoRAs. Distributed learning in that case would be the learning of the routing policy, but the data at the clients is not really used in learning the model parameters. Please correct me if I have misunderstood this, but you use the private data only to teach the router which experts to activate for which inputs? This requires some discussion and I would not take privacy here as the main focus on why you do this in FL. It needs a different justification then.
- **efficiency of router learning**: thank you for presenting a section on efficiency. I am not quite sure if this includes the learning of the router as well? Although the paper claims the router is small, FL still requires repeated router updates, importance score communication, expert quota updates. The impact of these components is not as deeply analyzed. Can you give more comments here?
- **evaluation is narrow**: my biggest concern is on the evaluation. While you give 3 models of different sizes, Llama-2 is quite outdated. I would expect to see a Llama-3 variant, especially as your method claims to be efficient. Also, the evaluation tasks are quite specialized and narrow. All tasks are skill-composition tasks involving Chinese and math/code. I think the chinese bias is too heavy here and I would have expected a more general set of tasks to evaluate on. Also, the tasks are exactly the same domains used to train the LoRAs. For example, the LoRA modules you use are: Chinese Chat LoRA, English Math LoRA, English Code LoRA. But then the tasks are: Chinese + Math, Chinese + Code, Math + Code. So the evaluation is basically: *compose LoRA modules trained for those exact domains and then test on tasks composed of those domains.* This is almost a best-case scenario for the method. Also, it would be good to evaluate on generic tasks as well, e.g., MMLU. This is cheap (you can use lm-eval-harness) and would strengthen the empirical claims.

Your results in the paper are good, but I would expect a more comprehensive evaluation to more standard tasks. If you claim that there are so many LoRA adapters on LoRAHub, then this should be not an issue to verify.

### **3. Miscellaneous**:

The overall presentation of the paper is good.
- Your writing style is clear and thorough.
- Your mathematical derivations are clear and easy to follow.
- Your figures are high-quality and easy to understand.

### **Summary**:
- I think more disucssion on the use case and privacy setting is needed.
- I also think that the current evaluation is a bit weak and would benefit from a newer model and more standard benchmarks/use cases other than chinese/math.

I am looking forward to your responses!

---

> ### Author Rebuttal · Authors · 2026-03-29
>
> We sincerely thank the reviewer for the positive and constructive feedback. Below are our point-by-point responses to your concerns.
>
> > **W1: Novelty.**
>
> Thank you for the insightful question.
> - **Differences:** Prior methods focus on centralized, module-level routing. SmartFed targets the computational and communication bottlenecks of edge devices, solving FL deployment barriers.
> - **Innovations:** SmartFed makes fundamental contributions:
>   - **Paradigm Shift:** Composing fully frozen, reusable LoRAs under strict FL constraints by optimizing only a lightweight router.
>   - **Fine-Grained Routing:** Resolving the coarse adaptation and poor scalability of module-level routing via rank-wise decomposition.
>   - **Adaptive Prioritization:** EEQA dynamically allocates the activation budget instead of using a static uniform quota, ensuring optimal resource utilization.
>
> > **W2: Baseline Adaptation.**
>
> We thank the reviewer for raising this important point.
> - **Fairness:** We construct faithful baselines from official implementations, preserving core architectures and applying FedAvg exclusively to learnable components under a strictly controlled, identical FL protocol.
> - **Reproducibility:** General FL hyperparameters (Appendix C) and baseline settings strictly match official codebases. For absolute transparency, the revision will add a detailed configuration table.
>
> > **W3&S1: Use Case Justification.**
>
> We appreciate the opportunity to clarify our conceptual positioning. SmartFed is driven by overcoming deployment bottlenecks, utilizing private data fundamentally differently:
> - **Efficiency Bottleneck:** Training LLMs (even LoRA) from scratch is prohibitively expensive. SmartFed shifts the FL objective to resource-efficient knowledge composition.
>
> - **Role of Private Data:**
>
>   - **Routing Policy:** Local data optimizes the router, teaching the model how to selectively assemble fine-grained experts based on target-domain semantics.
>
>   - **Allocation Policy:** Local data provides the importance feedback necessary to distribute the expert budget.
>
>     Without FL, learning these would necessitate centralizing private data.
>
> > **W4: Efficiency of Router Learning.**
>
> The evaluations encompass all overheads. We break down SmartFed’s efficiency at a granular level:
>
> - **Faster Convergence (Table A):** SmartFed converges significantly faster than FedIT because learning a routing policy is more efficient.
> - **Reduced Local Compute (Table B):** By computing gradients only for the router, SmartFed drastically reduces per-round wall-clock time.
> - **Reduced Communication (Table C):** Transmitting lightweight router and importance scores instead of LoRA significantly reduces communication overhead.
> - **Negligible Server Overhead:** EEQA quota updates execute via simple scalar arithmetic, incurring negligible latency to the system.
>
> **Table A: Number of Rounds for Convergence (MGSM).**
>
> | Method   | LLaMA2-7B | LLaMA2-13B |
> | -------- | --------- | ---------- |
> | FedIT    | 250       | 300        |
> | SmartFed | 20        | 20         |
>
> **Table B: Per-Round Computation Time (s) on H800 GPUs.**
>
> | Method   | LLaMA2-7B | LLaMA2-13B |
> | -------- | --------- | ---------- |
> | FedIT    | 105.3     | 197.9      |
> | SmartFed | 28.5      | 49.1       |
>
> **Table C: Per-Round Communication Cost (MB) Per Device.**
>
> | Method   | LLaMA2-7B | LLaMA2-13B |
> | -------- | --------- | ---------- |
> | FedIT    | 10.4      | 13.1       |
> | SmartFed | 0.8       | 1.0        |
>
>
> > **W5&S2: Breadth of Evaluation.**
>
> Thanks for the constructive comment.
>
> - **Task Generalization:** We integrate a general LoRA and evaluate on more tasks. SmartFed consistently outperforms FedIT, demonstrating strong generalization across diverse domains.
>
>   **Table D: Task Generalization Analysis (LLaMA2-7B).**
>
>   | Method   | MMLU      | BBH       | HumanEval |
>   | -------- | --------- | --------- | --------- |
>   | FedIT    | 42.45     | 39.28     | 17.59     |
>   | SmartFed | **51.84** | **46.61** | **21.28** |
>
> - **Model Generalization:** We further evaluate SmartFed on newer models. SmartFed maintains significant advantages across all models.
>
>   **Table E: Model Generalization Analysis.**
>
>   | Model     | Method   | MGSM      | DoIT      | GSM-Hard  | Average   |
>   | --------- | -------- | --------- | --------- | --------- | --------- |
>   | LLaMA3-3B | FedIT    | 35.42     | 41.48     | 48.20     | 41.70     |
>   |           | SmartFed | **46.89** | **53.57** | **60.21** | **53.56** |
>   | LLaMA3-8B | FedIT    | 47.55     | 56.14     | 64.92     | 56.20     |
>   |           | SmartFed | **61.28** | **70.33** | **78.44** | **70.02** |
>   | Qwen3-8B  | FedIT    | 50.25     | 58.54     | 66.49     | 58.43     |
>   |           | SmartFed | **64.21** | **72.99** | **79.56** | **72.25** |
>
> ---
>
> Finally, thanks again for your time and helpful comments. We hope your concerns are well addressed. For any other questions, we are happy to discuss them and make further clarifications.

---

> > ### Author Rebuttal · Reviewer_NsPn · 2026-03-31
> >
> > Thank you for the comprehensive rebuttal!
> >
> > Most of my points were resolved and the new results for the bigger/newer models solidify your method. I am still a bit unsure about the justification of the use case of your method, but it is becoming clearer now.
> >
> > I will consider raising my score!
> >
> > Thank you!

---

> > > ### Author Response · Authors · 2026-04-01
> > >
> > > Thank you for the acknowledgement of our work and for considering raising your score. We are thrilled by your positive feedback! We truly appreciate the time and effort you invested in reviewing our paper. We are glad our responses addressed your concerns, and we will include all newly added experiments and discussions in the final version.

---

### Official Review · Reviewer_wKrV · 2026-03-14

**Soundness:** 2
**Presentation:** 3
**Significance:** 3
**Originality:** 2
**Overall Recommendation:** 5
**Confidence:** 4

**Summary:**

This paper proposes SMARTFED, a federated fine-tuning framework for large language models that utilize existing LoRA modules instead of training new ones from scratch. The core idea is  decomposing LoRA modules into rank-wise experts (MoRE) and using a router to select relevant experts during inference. Additionally, the authors introduce Elastic Expert Quota Allocation (EEQA) to allocate computational budget based on expert importance. Experiments show that SMARTFED improves performance and training efficiency compared to existing LoRA composition and federated fine-tuning methods.

**Compliance With Llm Reviewing Policy:**

Affirmed.

**Final Justification:**

The authors have addressed most of my concerns and I raised my score.

**Key Questions For Authors:**

1. How does the method scale with many modules? How about the limitations such as storage and routing costs?
2. Could the authors provide a comprehensive ablation of what is mentioned in weaknesses?
3. What is the retrieval method used in the paper? Could the authors present some evaluations on the quality (such as noisy or not fully related), search cost, and latency of the retrieval?

**Limitations:**

The authors have included a statement without mentioning any impact or limitations. Authors could expand this section with discussion of communication, bias, privacy risks, and other technical constraints or impacts to strengthen it.

**Strengths And Weaknesses:**

Strengths:
1. The proposed method combines federated learning with parameter-efficient fine-tuning and mixture-of-experts to adapt LLMs on edge devices with reusing the existing knowledge.
2. The presented MoRE that is an extension of existing LoRA-composition, decomposes LoRA modules into rank-level experts, that enables fine-grained knowledge composition compared to existing approaches.
3. The framework trains only a lightweight router with frozen LoRA experts. This significantly reduces training and communication overhead in FL.

Weaknesses:
1. The paper assumes a task-relevant LoRA retrieval mechanism, but it never mentions how the initial retrieval is performed, nor evaluate the quality and cost of retrieval and its impact on performance. Furthermore, assuming existence of such high quality LoRA modules are not guaranteed in practice.
2. Ablation studies evaluate main components of the framework, though additional ablation on followings would strengthen the paper: number of retrieved modules (N), number of active experts (k), and robustness to quality of retrieved modules.
3. Experiments are based on a limited number of clients (20) The data distribution is also uniform, while the authors justify this due to absence of class labels, FL often involves strong heterogeneity such as domain-based splits.
4. Although Top K routing controls inference cost, the paper does not evaluate how routing and expert management scale when the number of modules increases largely.
5. The experiments are conducted on a few tasks and models. Evaluating the method on a broader range of tasks (such as QA, dialogue) would help better understand its general applicability.

---

> ### Author Rebuttal · Authors · 2026-03-29
>
> We sincerely thank the reviewer for the positive and constructive feedback. Below are our point-by-point responses to your concerns.
>
>
>
> > **W1&Q3: Retrieval Mechanism, Robustness, and LoRA Availability.**
>
> Thanks for the constructive comment.
>
> - **Retrieval Mechanism & Cost:** Devices can upload lightweight embeddings (or task descriptions) for server-side similarity matching. Since heavy search and matching are performed on the server, the overhead for devices is negligible.
>
> - **Robustness:** We evaluate on GSM-Hard by: 1) injecting a noisy LoRA into the expert pool; and 2) retrieving a weakly relevant general LoRA. Performance improves under noisy retrieval, and SmartFed still outperforms FedIT even with mismatched modules, confirming its robustness.
>
>   | Retrieved LoRA                | LLaMA2-7B | LLaMA2-13B |
>   | ----------------------------- | --------- | ---------- |
>   | Math + Code (Ideal)           | 64.19     | 72.65      |
>   | Math + Code + Chinese (Noise) | **64.25** | **72.83**  |
>   | Math + General (Mismatch)     | 58.97     | 69.02      |
>   | FedIT                         | 56.17     | 63.87      |
>
> - **Availability:** Task-specific LoRAs are abundant in the open-source ecosystem. Moreover, SmartFed does not require exact task-matched modules—only reasonably related reusable ones.
>
>
>
> > **W2&Q2: Ablation Study.**
>
> Thank you for the valuable comments.
>
> - **Retrieved Modules:** Performance improves as N increases on GSM-Hard. A broader pool provides a richer set of basis vectors, enabling the router to synthesize more precise adaptations.
>
>   | N    | LLaMA2-7B | LLaMA2-13B |
>   | ---- | --------- | ---------- |
>   | 2    | 64.19     | 72.65      |
>   | 3    | 66.43     | 75.49      |
>   | 4    | 67.49     | 76.89      |
>
> - **Active Experts:** Increasing the active expert quota steadily enhances accuracy on GSM-Hard. Activating more experts increases the expressiveness of the composed update, capturing finer-grained knowledge.
>
>   | k    | LLaMA2-7B | LLaMA2-13B |
>   | ---- | --------- | ---------- |
>   | 16   | 58.42     | 68.04      |
>   | 32   | 64.19     | 72.65      |
>   | 64   | 68.92     | 77.51      |
>
> - **Robustness:** Please refer to W1.
>
> > **W3: Client Scale and Data Heterogeneity.**
>
> We evaluate scalability by expanding DoIT to 100 clients and simulate Non-IID through task-domain partitioning (α=0.5). SmartFed consistently outperforms FedIT on LLaMA2-7B. Notably, SmartFed's performance under Non-IID (53.04) surpasses FedIT's IID performance (46.25). This demonstrates that SmartFed provides a higher performance ceiling, guaranteeing high utility despite highly skewed local data.
>
> | Distribution | FedIT | SmartFed  |
> | ------------ | ----- | --------- |
> | IID          | 46.25 | **56.81** |
> | Non-IID      | 44.72 | **53.04** |
>
> > **W4&Q1: Scalability of Routing and Expert Management.**
>
> Thanks for the insightful question.
>
> - **Routing Cost:** Although retrieving more modules enlarges the router, its lightweight linear projections keep computational costs negligible. Routing 16 modules adds merely **0.72%** to the base model's FLOPs (LLaMA2-7B).
>
> - **Management Cost:** Because experts are frozen, they incur zero gradient-memory overhead. Storing 16 LoRAs requires only 80 MB, ensuring negligible storage footprint.
>
>   | Metric | Base Model | N=4        | N=8        | N=16       |
>   | ------ | ---------- | ---------- | ---------- | ---------- |
>   | TFLOPs | 2228       | 4 (0.18%)  | 8 (0.36%)  | 16 (0.72%) |
>   | MB     | 27574      | 20 (0.07%) | 40 (0.15%) | 80 (0.29%) |
>
>
> > **W5: Performance Across Broader Tasks and Models.**
>
> Thanks for the constructive comment.
>
> - **Task Generalization:** We evaluate SmartFed on broader tasks (LLaMA2-7B) by integrating a general LoRA. SmartFed consistently outperforms FedIT, demonstrating strong generalization across diverse domains.
>
>   | Method   | MMLU      | BBH       | HumanEval |
>   | -------- | --------- | --------- | --------- |
>   | FedIT    | 42.45     | 39.28     | 17.59     |
>   | SmartFed | **51.84** | **46.61** | **21.28** |
>
> - **Model Generalization:** We further evaluate SmartFed on newer models. SmartFed maintains significant advantages over FedIT, confirming it as a robust, model-agnostic framework.
>
>   | Model     | Method   | MGSM      | DoIT      | GSM-Hard  | Average   |
>   | --------- | -------- | --------- | --------- | --------- | --------- |
>   | LLaMA3-8B | FedIT    | 47.55     | 56.14     | 64.92     | 56.20     |
>   |           | SmartFed | **61.28** | **70.33** | **78.44** | **70.02** |
>   | Qwen3-8B  | FedIT    | 50.25     | 58.54     | 66.49     | 58.43     |
>   |           | SmartFed | **64.21** | **72.99** | **79.56** | **72.25** |
>
> > **L1: Impact Statement.**
>
> We will expand the Impact Statement to explicitly discuss these critical dimensions.
>
> ---
>
> Finally, thanks again for your time and helpful comments. We hope your concerns are well addressed. For any other questions, we are happy to discuss them and make further clarifications.

---

> > ### Author Rebuttal · Reviewer_wKrV · 2026-04-03
> >
> > The authors have addressed most of my concerns and I will raise my score.

---

> > > ### Author Response · Authors · 2026-04-04
> > >
> > > Thank you for the acknowledgement of our work and for your intention to raise the score! We would greatly appreciate it if you could kindly reflect this score update in the review system. We truly appreciate the time and effort you invested in reviewing our paper. We are glad our rebuttal has adequately addressed your concerns, and we will ensure all newly added experiments, clarifications, and discussions are included in the final version.

---

### Decision · Program_Chairs · 2026-04-30

**Decision:**

Accept (spotlight)

**Comment:**

This paper proposes SmartFed, a framework for resource-efficient federated fine-tuning of Large Language Models (LLMs) by reusing existing LoRA modules rather than training from scratch. The core technical contributions include the Mixture of Rank-Wise Experts (MoRE), which decomposes LoRA modules into fine-grained rank-level experts for selective activation, and Elastic Expert Quota Allocation (EEQA), which adaptively distributes computational budgets based on expert importance.

The reviewers generally agree that the problem setting—adapting LLMs on resource-constrained edge devices via knowledge reuse—is timely and significant. The proposed decomposition of LoRA modules into rank-wise experts is viewed as a novel and intuitive refinement over existing module-level merging or routing approaches. Initial reviews highlighted concerns regarding the limited scope of evaluation (older model architectures, narrow task domains, small client scale), the lack of clarity on the LoRA retrieval mechanism, and underdeveloped discussions on privacy and robustness to noisy experts.

During the rebuttal phase, the authors provided extensive additional experiments that effectively addressed these concerns. Specifically, they: Expanded evaluations to newer model families (LLaMA-3, Qwen-3) and smaller scales (3B), demonstrating generalizability. Tested scalability with up to 100 clients under Non-IID data distributions, showing robust performance gains over baselines. Provided empirical evidence on robustness against noisy or mismatched retrieved LoRAs and analyzed the overhead of the retrieval and routing mechanisms.
Included results under Differential Privacy settings and broadened the benchmark suite to include general tasks (MMLU, BBH, HumanEval).

All reviewers acknowledged the thoroughness of the rebuttal and raised their scores accordingly (ranging from Weak Accept to Strong Accept). While some minor reservations remain regarding the precise practical deployment scenarios for "public LoRA merging" in private FL settings, the consensus is that the technical soundness, empirical validation, and efficiency gains presented make this a solid contribution. The method offers a clear pathway for reducing communication and computation costs in federated LLM adaptation.